# Immobilization of Silver Nanoparticles with Defensive Gum of *Moringa oleifera* for Antibacterial Efficacy Against Resistant Bacterial Species from Human Infections

**DOI:** 10.3390/ph17111546

**Published:** 2024-11-18

**Authors:** Liaqat Ali, Nisar Ahmad, Muhammad Nazir Uddin, Mostafa A. Abdel-Maksoud, Hina Fazal, Sabiha Fatima, Mohamed A. El-Tayeb, Bushra Hafeez Kiani, Wajid Khan, Murad Ali Rahat, Mohammad Ali, Yaqub Khan, Kamran Rauf, Salman Khan, Sami Ullah, Tanveer Ahmad, Afshan Salam, Sajjad Ahmad

**Affiliations:** 1Centre for Biotechnology and Microbiology, University of Swat, Swat 19120, Pakistan; liaqatbiotech23@gmail.com (L.A.); nazir@uswat.edu.pk (M.N.U.); w.khan@uswat.edu.pk (W.K.); muradrahat@uswat.edu.pk (M.A.R.); alimoh@uswat.edu.pk (M.A.); shaazkhan068@gmail.com (Y.K.); salmanbiotech6168@gmail.com (S.K.); samibiotech22@gmail.com (S.U.); tanveerbiotech88@gmail.com (T.A.); awaisnaveed085@gmail.com (A.S.); sajjad.std.mus@gmail.com (S.A.); 2Botany and Microbiology Department, College of Science, King Saud University, P.O. Box 2455, Riyadh 11451, Saudi Arabia; mabdmaksoud@ksu.edu.sa; 3Pakistan Council for Scientific and Industrial Research (PCSIR) Laboratories Complex, Peshawar 25120, Pakistan; 4Department of Clinical Laboratory Science, College of Applied Medical Sciences, King Saud University, P.O. Box 10219, Riyadh 12371, Saudi Arabia; sabmehdi@ksu.edu.sa (S.F.); mali5@ksu.edu.sa (M.A.E.-T.); 5Department of Biology and Biotechnology, Worcester Polytechnic Institute, Worcester, MA 01609, USA; bushra.hafeez@iiu.edu.pk; 6Department of Horticulture, The University of Agriculture Peshawar, Peshawar 25120, Pakistan; raufkamran317@gmail.com

**Keywords:** *Moringa oleifera*, organic synthesis of silver nanoparticles, defensive gum, immobilization of AgNPs, antibiotic resistance, antibacterial activity, alternative antimicrobials

## Abstract

**Background**: The worldwide misuse of antibiotics is one of the main factors in microbial resistance that is a serious threat worldwide. Alternative strategies are needed to overcome this issue. **Objectives**: In this study, a novel strategy was adopted to suppress the growth of resistant pathogens through immobilization of silver nanoparticles (AgNPs) in gum of *Moringa oleifera*. **Methods**: The AgNPs were prepared from the leaves of *Moringa oleifera* and subsequently characterized through UV-spectrophotometry, FTIR, SEM, and XRD. The differential ratios of characterized AgNPs were immobilized with gum of *M. oleifera* and investigated for antimicrobial potential against highly resistant pathogens. **Results**: The immobilized AgNPs displayed promising activities against highly resistant *B. subtilis* (23.6 mm; 50 µL:200 µL), *E. coli* (19.3 mm; 75 µL:200 µL), *K. pneumoniae* (22 mm; 200 µL:200 µL), *P. mirabilis* (16.3 mm; 100 µL:200 µL), *P. aeruginosa* (22 mm; 175 µL:200 µL), and *S. typhi* (19.3; 25 µL:200 µL) than either AgNPs alone or gum. The immobilized AgNPs released positive sliver ions that easily attached to negatively charged bacterial cells. After attachment and permeation to bacterial cells, the immobilized NPs alter the cell membrane permeability, protein/enzymes denaturation, oxidative stress (ROS), damage DNA, and change the gene expression level. It has been mechanistically considered that the immobilized AgNPs can kill bacteria by damaging their cell membranes, dephosphorylating tyrosine residues during their signal transduction pathways, inducing cell apoptosis, rupturing organelles, and inhibiting cell division, which finally leads to cell death. **Conclusions**: This study proposes a potential alternative drug for curing various infections.

## 1. Introduction

*Moringa oleifera* (*M. oleifera*), the “Miracle tree”, thrives globally in almost all tropical and subtropical regions, but it is believed to be native to India, Bangladesh, Pakistan and Afghanistan [1]. The *Moringa* genus, belonging to the *Moringaceae* family, is also known as the drumstick tree, horseradish tree and benzoyl tree and can grow in a wide range of environmental conditions [2,3,4,5,6,7]. Moreover, *M. oleifera* is a rich source of different phytochemicals [8,9,10,11]. *M. oleifera* is a highly nutritional plant, and its various anatomical structures have been used for centuries to treat a variety of health conditions and possess widespread biological properties [12,13,14,15,16,17].

*M. oleifera* has been shown to have antimicrobial and antifungal activity. The plant contains compounds that can inhibit the growth of bacteria such as *Pseudomonas aeruginosa*, *Salmonella typhi*, *Vibrio cholera*, *Escherichia coli*, *Staphylococcus aureus*, *Klebsiella pneumoniae* and *Candida albicans* [18,19], and fungi such as *Rhizopus stolonifera*, *Neurospora crassa*, *Aspergillus niger*, and *Microsporum gypseum* [20], including pterygospermin and ethanolic extracts of the leaves, and alkaloids, flavonoids, and steroids in the fruits [19], *M. oleifera* has been shown to have antioxidant properties. The plant contains compounds that can scavenge free radicals [21], such as kaempferol [22], myricetin, isoquercetin, astragalin, and crypto-chlorogenic acid [23]. This antioxidant potential can help to protect cells from damage and reduce the risk of chronic diseases.

*M. oleifera* bark extract has anti-inflammatory activity similar to diclofenac and is thought to work by regulating the c-Jun N-terminal kinase and neutrophil pathways [24,25] due to the presence of flavonoids, alkaloids, vanillin, moringin, phenols, tannins, carotenoids, hydroxymellein, Beta-sitosterol, and ß-sitostenol [26]. *M. oleifera* has been shown to have both fertility and anti-fertility properties. The aqueous extract of the plant has been found to increase abortifacient and anti-fertility at doses of 200 and 400 mg/kg [15]. A recent study suggested that the ingestion of *M. oleifera* before, after and during pregnancy may lead to adverse fetal developmental outcomes [27].

The seed extract was able to reduce histamine release and suppress anaphylaxis induced by anti-immunoglobulin G. The mechanism underlying this effect is thought to be the membrane-stabilizing potential of the extract on mast cells [28]. The leaf extract can prevent gastric ulcers by reducing free radicals, neutralizing stomach acid, and increasing capillary resistance [29,30]. Some active constituents such as niazirmin A, niazirmin B, and niazimicin play a key role in lowering cholesterol levels, and avenasterol, stigmasterol, campesterol, and beta-sistosterol have diuretic activity [28,31,32]. The presence of quercetin, kaempferol, flavonoids, and benzylglucosinolate can protect the liver from damage [33,34]. The leaves, seeds, and dried pulp have shown health-inducing properties [28,35,36]. It is a nutrient-rich plant with a high protein content, as well as significant amounts of vitamins, minerals, fat carbohydrates, and dietary fibers [37,38,39].

Plant exudates are a diverse group of compounds that can be released from all parts of the plant [40]. Gums are complex carbohydrates that can absorb water and form gels [41]. Plant gums are bioavailable and have been used for centuries in a variety of applications, including food, medicine, and industry [41,42]. In pharmaceutical formulations, plant-based gums and mucilages are the key ingredients due to their bootability, widespread availability, non-toxicity, and affordability [43]. Scientists and pharmaceutical companies are interested in using plant-based gums and mucilages as drug delivery systems, gene therapy vectors, and biosynthetic materials. By modifying these natural substances, researchers have developed a new class of innovative drug products with potential applications in a variety of advanced fields [41]. Plant-based gums are non-toxic and biodegradable [41,44]. They have shown antimicrobial activity against a variety of microorganisms, including bacteria, fungi, and viruses [45]. Potential mechanisms of action include the inhibition of the bacterial cell membrane, chelation of essential metal ions, production of reactive oxygen species, and modulation of the host immune response [46,47]. Plant-based gums could be used to develop a variety of antimicrobial products, such as food additives, pharmaceutical products, and agricultural products [47].

Nanomaterials are at the forefront of nanotechnology, a rapidly developing field with the potential to revolutionize many areas of human activity [48]. Nanoparticles are particles with a maximum size of 100 nanometers (nm), which is about 1/100,000th the width of a human hair [49]. Nanoparticles exhibit unique properties that make them superior and indispensable in many applications, including combating microbes, drug delivery, catalysis, water purification, the treatment of environmental waste, in the food industry, in the textile industry, biolabeling, and cancer treatment [50]. Silver nanoparticles (AgNPs) are a promising antimicrobial agent because they are effective against a wide range of microorganisms, including bacteria, fungi, and viruses [51,52]. They are also stable and reduce the likelihood of bacteria developing antibiotic resistance. AgNPs are being investigated for use in a variety of applications, such as medicine, medical devices, and food packaging [53,54].

Immobilization is a powerful technique that allows for reuse and recovery [55], improved stability [56], controlled reactivity, and easier separation of molecules [57]. It is used in a wide range of applications, including bio-catalysis [58], sensors [59], diagnostics [60], and environmental remediation [61]. The immobilization of AgNPs is a versatile technique that can be used to improve their stability, control their reactivity, and make them easier to use in a variety of applications: AgNPs have strong antimicrobial properties and can be used to create coatings that can kill bacteria and other microorganisms [62], AgNPs can be used to create sensors that are sensitive to specific chemicals or biological signals, AgNPs can be used as catalysts for a variety of chemical reactions, and AgNPs can be used to degrade pollutants in water and soil [63].

AgNPs alone have proven antimicrobial potential, while the gum is released in response to pathogen attacks in plants. Here, the dual and synergistic potential of AgNPs along with the gum was investigated for the growth inhibition of infectious strains. Immobilized AgNPs have potential to kill infectious strains by disrupting the membranes by inducing pores and causing cytoplasmic leakage and bacterial cell death. It is possible that immobilized AgNPs can stimulate the defense system of bacterial cells more rapidly than either NPs alone or using gum alone. The immobilization strategy most probably releases huge quantities of reactive oxygen species or other radicals that directly react and inhibit major functions such as metabolism, replication, translation, and denaturation and cause DNA damage.

The gum of this miracle plant possesses strong antimicrobial potential against various pathogens. Similarly, silver nanoparticles are widely used in the medical sector due to their high inhibiting efficacy against microorganisms. Therefore, the main objective of the current study was to investigate the synergistic and high efficacy of immobilized silver nanoparticles encapsulated in *M. oleifera* gum against highly resistant microbes and to develop a possible antimicrobial mechanism for the immobilized AgNPs.

## 2. Results

### 2.1. Biogenic Synthesis of Nanoparticles and Its Characterization

*M. Oleifera* leaves were mixed with different volumes of salt solution, keeping the reactant volume constant. The change in color representing the formation of NPs was observed in two different ways. Upon exposing the reactants to sunlight or incubation for 24 h, the observation showed that a 4 mg/mL:4 mM ratio displayed the darkest color and characteristic peak between 300 and 600 nm. The other differential ratios were less effective in the formation of NPs.

#### 2.1.1. UV–Visible Spectrophotometry of Leaf-Derived AgNPs

The initial step in identifying the formation of silver nanoparticles from leaf extracts involved using an Elite Double-beam UV spectrophotometer to measure the absorbance at 300–600 nm for 24 h. The reactants (4 mg/mL:4 mM) showed the greatest absorbance in the 400–450 nm range, with a peak at 430 nm. Furthermore, the solution’s color changed from light brown to dark brown, possibly indicating the formation of nanoparticles as shown in Figure 1.

#### 2.1.2. Fourier Transform Infrared Spectroscopy (FTIR) ML-Derived AgNPs

FTIR was used to analyze the chemicals in the silver nanoparticles. The FTIR spectra displayed different plant substances (phenolics functional group), like proteins and alcohols/carbonyl rings/groups, and aromatic amines and this all helped the nanoparticles. X-rays showed that the nanoparticles were round-shaped and about 25 to 40 nanometers in size (Figure 2). The broad peak was observed at 2000, corresponding to carboxylic acids with an OH group with alcohol stretching. The peak at 3000 to 2900 represents the major intramolecular bonds with OH groups. The C-H of Aldehydic amines was represented by the spectra with a peak area of 2800 with C-H stretching, while after the sp3 hybridized groups, the O=C=O displayed a peak area ranging from 2400 to 2100. Below the 1700 peak area, double carbon bonding was observed, showing the presence of alkenes. N alone or O alone or in various groups either in nitro compounds in FTIR spectra was observed at the peak area below 1500, while the OH of the amine and phenolic compounds was observed from 1300 to 1450. Peaks below 650 cm^−1^ were attributed to the AgNPs (Figure 2).

#### 2.1.3. X-Ray Diffraction Spectrum (XRD) Analysis of Biogenic AgNPs

X-ray diffraction (XRD) analysis of the biogenic silver nanoparticles synthesized from *M. oleifera* leaf extract revealed a mostly round/cubic crystalline structure with three strong Bragg reflections at 32.201, 37.27, and 45.6, corresponding to the (42), (40), and (120) planes, respectively. The interplanar distances were calculated as 2.15 Å, 2.11 Å, and 1.38 Å, respectively. The particle size, determined by the Debye–Scherrer formula, was 25 nm (Figure 3). These findings confirm the characteristics of silver nanoparticles derived from *Moringa oleifera*. The XRD analysis was conducted using a diffractometer with Cu K-alpha radiation at 40 kV and 30 mA. The scanning range was from 0 to 80 degrees 2θ with a scan rate of 10 degrees/min. The XRD analysis was performed using a Goniometer PW3050/60 and analyzed with Xpert-PRO software (https://www.xpert-pro.com/home.php, (accessed on 29 October 2024)).

#### 2.1.4. Transmission Electron Microscopy of ML-Derived AgNPs

The ML-derived AgNPs were analyzed by Transmission Electron Microscopy (TEM) to determine their size, shape, and morphology. The TEM images displayed that most of the ML-derived AgNPs were spherical in shape, and they may be individual or in aggregated morphologies (Figure 4). The presence of bioactive phytochemicals in the leaves of medicinally important *M. oleifera* extract was the main characteristic responsible for the formation of spherical-shape nanoparticles. The scale on the TEM images indicates that the size varies from 25 nm to 40 nm to 120 nm, as shown in Figure 4.

### 2.2. Morphological Identification of Bacterial Isolates

To identify bacterial species, morphological identification was carried out. *E. coli* on MAC agar has a bright, smooth, pink-colored elevated colony that is transparent; *K. pneumoniae* on MAC agar revealed the development of small to large-sized mucoid colonies with a creamy pink and yellow appearance, as shown in Figure 5b; *P. aeruginosa* on CLED agar showed green colonies with a typical matte surface and rough periphery—its scent was “sweet” and it was colored blue-green, as shown in Figure 5c. *S. typhi* on MAC agar appeared pale, colorless, smooth, and transparent, with raised colonies, as shown in Figure 5d.

### 2.3. Identification Through Microscopy

Morphology and staining characteristics of *E. coli*, *K. pneumoniae*, *P. aeruginosa*, and *S. typhi* were observed under microscope. *E. coli* was observed as Gram-negative, rod-shaped, pink, and grouped in single or paired/short, as shown in Figure 6a. *K. pneumoniae*, was bacilli in shape and Gram-negative, as shown in Figure 6b. *P. aeruginosa* was rod-shaped, Gram-negative bacteria, typically 0.5–1.0 µm in width and 1–4 µm in length, as shown in Figure 6c, and *S. typhi* was short, singly arranged, Gram-negative rod-shaped bacteria, as shown in Figure 6d.

### 2.4. Antibiotic Susceptibility Pattern

The sensitivity patterns of *B. subtilis*, *E. coli*, *K. pneumoniae*, *P. aeruginosa*, *P. mirabilis*, *and S. typhi* isolates were investigated using the disk diffusion technique. Doxycycline (DO), Ceftriaxone (CRO), Cefotaxime (CTX), Sulbactam/cefoperazone (SCF), Erythromycin (E), Trimethoprim/sulfamethoxazole (SXT), Imipenem (IPM), and Azithromycin (AZM) were applied on each bacterial strain for a susceptibility test, as shown in Table 1 and Figure 7. *B. subtilis* was totally resistant to E, SXT, and AZM. *E. coli* displayed resistance to SCF, E, and AZM. *K. pneumoniae* showed resistance to CRO, E, and SXT. *P. aeruginosa* exhibited resistance to CRO, CTX, SCF, E, and SXT. *P. mirabilis* was resistant to CRO, SXT, and AZM. *S. typhi* was resistant to CRO, SXT, and AZM.

### 2.5. Antibacterial Potential of Immobilized AgNPs

Many microorganisms that cause human diseases are becoming resistant to antibiotics due to the misuse of antibiotics. This means that there is a need to find novel antibacterial substances derived from plants and other natural sources. To find traditionally utilized therapeutic herbs that are very effective against both Gram-positive and Gram-negative bacteria, different concentrations of *M. oleifera* gums (gums alone, gums + AgNPs, and AgNPs alone) were applied to investigate antibacterial activities. The synergistic combination boosted the antibacterial efficacy against resistant bacteria. Furthermore, several secondary metabolites, including alkaloids, flavonoids, glycosides, phenols, saponins, and sterols, are some of the active phytochemicals that give medicinal plants their therapeutic qualities. Various plant extracts, such as of the root, stem, bark, and seeds, have been shown to contain these metabolites. Therefore, preliminary screening tests are useful for finding bioactive compounds that could result in their finding to launch novel pharmaceuticals. Additionally, these tests can help to quantify and qualitatively separate chemicals with pharmacological action [64]. Both the AgNPs and gum of *M. oleifera* have high antimicrobial potential, and their immobilization further enhances the antibacterial capability by inhibiting the growth of resistant bacterial strains. The immobilization of AgNPs and gum is presented in the following Figure 8.

### 2.6. Immobilized AgNP-Induced Growth Inhibition of Highly Resistant B. subtilis

The zone of inhibition of the synergistic combination of *M. oleifera* gum and AgNPs against highly resistant *B. subtilis* displayed variable activities: T1 (25 µL AgNPs:200 µL gums) showed a 19.3 mm zone inhibition; T2 (50 µL AgNPs:200 µL gums), 23.6 mm; T3 (75 µL AgNPs:200 µL gums), 20.6 mm; T4 (100 µL AgNPs:200 µL gums), 17 mm; T5 (125 µL AgNPs:200 µL gums), 15.6 mm; T6 (150 µL AgNPs:200 µL gums), 14.6 mm; T7 (175 µL AgNPs:200 µL gums), 16 mm; T8 (200 µL AgNPs:200 µL gums), 21.3 mm; T9 (225 µL AgNPs:200 µL gums), 19.3 mm; and T10 (250 µL AgNPs:200 µL gums), 11.6 mm. This mean that these combinations of immobilized AgNPs were able to kill bacteria within a radius of 11.6 to 23.6 mm. AgNPs alone as C1 (100 µL AgNPs) and gums alone as 200 µL gums showed zones of inhibition of 16.3 mm and 19 mm, respectively. These results suggest that the immobilized AgNPs were more effective at killing highly resistant *B. subtilis* than either AgNPs or gums alone, as shown in Figure 9a. The zones of inhibition of each treatment are shown in Figure 10.

### 2.7. Immobilized AgNP-Induced Growth Inhibition of Highly Resistant E. coli

The immobilized AgNPs also exhibited excellent inhibiting potential against highly resistant *E. coli* and displayed a 17.3 mm zone using T1 (25 µL AgNPs:200 µL gums), while, the other treatments, such as T2 (50 µL AgNPs:200 µL gums), showed a 17.6 mm zone; T3 (75 µL AgNPs:200 µL gums), 19.3 mm; T4 (100 µL AgNPs:200 µL gums), 15 mm; T5 (125 µL AgNPs:200 µL gums), 18.3 mm; T6 (150 µL AgNPs:200 µL gums), 15.3 mm; T7 (175 µL AgNPs:200 µL gums), 17.6 mm; T8 (200 µL AgNPs:200 µL gums), 14.3 mm; T9 (225 µL AgNPs:200 µL gums), 17.3 mm; and T10 (250 µL AgNPs:200 µL gums), 17 mm. By comparison, this was more than C1 and C2. This means that the immobilized AgNPs were found to be dominant in killing resistant *E. coli* compared to other treatments (Figure 9b), and the zones of inhibitions are given in Figure 11.

### 2.8. Immobilized AgNP-Induced Inhibition of Highly Resistant K. pneumoniae

In this study, the immobilized AgNPs presented potent activities against resistant *K. pneumoniae* as T1 (25 µL AgNPs:200 µL gums), 16.6 mm; T2 (25 µL AgNPs:200 µL gums), 12.6 mm; T3 (75 µL AgNPs:200 µL gums), 18.3 mm; T4 (100 µL AgNPs:200 µL gums), 17.3 mm; T5 (125 µL AgNPs:200 µL gums), 13.6 mm; T6 (150 µL AgNPs:200 µL gums), 15.3 mm; T7 (175 µL AgNPs:200 µL gums), 19 mm; T8 (200 µL AgNPs:200 µL gums), 22 mm; T9 (225 µL AgNPs:200 µL gums), 19 mm; and T10 (250 µL AgNPs:200 µL gums), 17.6 mm. The C1 and C2 displayed lower activities than the immobilized AgNPs. The inhibiting efficacy against highly resistant *K. pneumoniae* is shown in Figure 9c and Figure 12.

### 2.9. Immobilized AgNP-Induced Growth Inhibition of Highly Resistant P. mirabilis and the Zones of Inhibition of the Synergistic Combination of M. oleifera Gums

AgNPs (immobilized) against highly resistant *P. mirabilis* were recorded as follows: T1 (25 µL AgNPs:200 µL gums) displayed 13.3 mm; T2 (50 µL AgNPs:200 µL gums), 16 mm; T3 (75 µL AgNPs:200 µL gums), 14.3 mm; T4 (100 µL AgNPs:200 µL gums), 16.3 mm; T5 (125 µL AgNPs:200 µL gums), 14.3 mm; T6 (150 µL AgNPs:200 µL gums), 16 mm; T7 (175 µL AgNPs:200 µL gums), 11.6 mm; T8 (200 µL AgNPs:200 µL gums), 13.6 mm; T9 (225 µL AgNPs:200 µL gums), 16 mm; and T10 (250 µL AgNPs:200 µL gums) 15 mm. However, the C1 (100 µL AgNPs) and gums alone as C2 (200 µL gums) have shown 16 mm and 13 mm zones of inhibition, respectively. These results confirm that the immobilized nanoparticles were superior in efficacy compared to the AgNPs or gum of medicinal plants used alone against highly resistant *P. mirabilis*, as shown in Figure 9d and Figure 13.

### 2.10. Immobilized AgNP-Induced Inhibition of Highly Resistant P. aeruginosa

The immobilized AgNPs exhibited optimal activities against the highly resistant *P. aeruginosa*. The coating of NPs with gum enhanced the antibacterial effectiveness, as shown in Figure 9e and Figure 14. The differential treatments of immobilized AgNPs exhibited differential activities against *P. aeruginosa* and the activities in terms of inhibition zones were T1 (25 µL AgNPs:200 µL gums), 18 mm; T2 (50 µL AgNPs:200 µL gums), 20.6 mm; T3 (75 µL AgNPs:200 µL gums), 16.3 mm; T4 (100 µL AgNPs:200 µL gums), 19 mm; T5 (125 µL AgNPs:200 µL gums), 21 mm; T6 (150 µL AgNPs:200 µL gums), 22 mm; T7 (175 µL AgNPs:200 µL gums), 20 mm; T8 (200 µL AgNPs:200 µL gums), 22 mm; T9 (225 µL AgNPs:200 µL gums), 17.6 mm; and T10 (250 µL AgNPs:200 µL gums), 14 mm. The AgNPs alone, C1 (100 µL AgNPs), and gums alone, C2 (200 µL gums), exhibited 17 mm and 16 mm zones of inhibition, which are comparatively less than immobilized AgNPs and more effective in killing the highly resistant *P. aeruginosa*, as shown in Figure 9e and Figure 14.

### 2.11. Immobilized AgNP-Induced Growth Inhibition of S. typhi

Herewith, the zones of inhibition of the immobilized AgNPs against highly resistant *S. typhi* were recorded as T1 (25 µL AgNPs:200 µL gums), 19.3 mm; T2 (50 µL AgNPs:200 µL gums), 15.6 mm; T3 (75 µL AgNPs:200 µL gums), 19 mm; T4 (100 µL AgNPs:200 µL gums), 14 mm; T5 (125 µL AgNPs:200 µL gums), 14.6 mm; T6 (150 µL AgNPs:200 µL gums), 12.6 mm; T7 (175 µL AgNPs:200 µL gums), 13.6 mm; T8 (200 µL AgNPs:200 µL gums), 14.6 mm; T9 (225 µL AgNPs:200 µL gums), 15.6 mm; and T10 (250 µL AgNPs:200 µL gums), 16.6 mm. The differential ratios of immobilized AgNPs displayed antibacterial potential within the range of 12.6 to 19.3 mm, which was significantly slightly higher than that of C1 (100 µL AgNPs) and C2 (200 µL gums), which exhibited 14.3 mm and 16.3 mm zones of inhibition, respectively. This means that the immobilized AgNPs were also more effective than NPs or gum alone against resistant *S. typhi*, as shown in Figure 9f and Figure 15.

## 3. Discussion

Plant gums are natural substances that are not harmful to humans or to the environment. They are also sustainable, meaning that they can be produced and used without depleting natural resources. Additionally, they are recyclable, cost-effective, and biodegradable. These properties make them ideal for use in a variety of industries, including pharmaceuticals, nanoparticle synthesis, and food. *M. oleifera* is a highly important medicinal plant that is found in both tropical and subtropical regions around the globe. Numerous significant phenols, amino acids, proteins, vitamins, and betacarotenes may be found in all anatomical structures of the plant. Antiulcer, antidiuretic, anticancer, antipyretic, anti-inflammatory, antidiabetic, antihypertensive, antidiabetic, cholesterol-lowering, and antioxidant qualities are found in all parts of this plant. Traditional medical systems have traditionally used this herb since ancient times, particularly in south Asia [65].

Multidrug-resistant (MDR) bacteria are extremely dangerous and pose a serious threat to global health systems because they can survive antibiotic treatments. These bacteria are able to adapt to antibiotics by changing their genetic makeup to become resistant to known antibiotics and their combinations [66]. Nanoparticles are a new type of antimicrobial agent that are being researched as a way to combat these MDR bacteria; they target the bacterial cells in multiple ways, making it difficult for the bacteria to escape [67]. The emergence of antibiotic-resistant bacteria has made it difficult to treat bacterial infections. There is an urgent need for new and powerful antibacterial agents. AgNPs have shown excellent antibacterial activity. AgNPs can target bacterial cells in multiple ways, such as by altering cell membrane permeability and protein denaturation, causing oxidative stress, deactivating enzymes, generating ROS, damaging DNA, and changing gene expression. These unique properties make it difficult for bacteria to develop resistance to AgNPs.

In this study, immobilized AgNPs were effective against both Gram-positive and Gram-negative MDR bacteria. These synergistic combinations showed exceptional activity against *B. subtilis* in T2 (50 µL AgNPs:200 µL gums) with a zone of inhibition of 23.66 mm. They also showed excellent activity against *K. pneumoniae* in T8 (200 µL:200 µL) with a zone of inhibition of 22 mm and good activity against *P. aeruginosa* in T6 (150 µL:200 µL) with a zone of inhibition of 22 mm. Against *E. coli*, T3 (75 µL:200 µL) displayed a zone of inhibition of 19.33 mm, and 19.33 mm against *S. typhi* as T1. Additionally, they showed activity against *P. mirabilis* in T4 (100 µL:200 µL) with zone of inhibition of 16.33 mm. The authors of [68] reported that the antimicrobial activity of AgNPs against *K. pneumonia* was 14.33 mm, 15.66 mm against *P. aeruginosa*, 15.33 mm against *E. coli*, and 13.33 mm against *P. mirabilis*, and further reported that the antimicrobial activity of streptomycin against *K. pneumoniae* was 16.33 mm, 16.0 mm against *P. aeruginosa*, 16.0 mm against *E. coli*, and 15.33 mm against *P. mirabilis*. They further investigated the antimicrobial activity of the combination of AgNPs and streptomycin against *K. pneumoniae* (19.66 mm), *P. aeruginosa* (20.0 mm), *E. coli* (20.3 mm), and *P. mirabilis* (17.66 mm). The authors of [69] reported that the antimicrobial activity of levofloxacin against *S. typhi* was 15.0 mm and the combination of levofloxacin and AgNPs possesses antimicrobial activity against *S. typhi* with a 20 mm zone of inhibition.

*M. oleifera* has been shown to have antifungal, antibacterial, and water-purification properties. The compound pterygospermin found in the roots and flowers of the plant has antifungal activity against *Neurospora crassa*, *Rhizopus stolonifera*, *Microsporum gypseum*, and *Aspergillus niger* [20]. A range of microbes, including *P. aeruginosa*, *Vibrio cholera*, and *S. typhi*, are resistant to the ethanolic extract of the leaves [18]. The seed extracts of the plants have been demonstrated to inhibit the growth of bacteria in both liquid and on solid media, suggesting that they could be used in water purification [70]. Methanol extracts of the leaves have also been shown to repress microbes such as *K. pneumoniae* and *Staphylococcus aureus* of the urinary tract infections. Alkaloids, flavonoids, and steroids found in *M. oleifera* fruits have an inhibiting impact on Candida albicans cultures either through protein denaturation or by residing in the spore’s germination through their steroid ring [19].

In this study, the immobilized AgNPs mechanistically cause the cell death of resistant bacterial strains by disrupting the cell membrane, leading to the leakage of cellular contents and then cell death. The immobilized AgNPs generated reactive oxygen species (ROS), which damage cellular components, such as proteins, DNA, and lipids. In addition, immobilized AgNPs denature bacterial proteins, disrupting important cellular processes such as metabolism, DNA replication, and protein synthesis. The immobilized AgNPs kill bacterial cells through a combination of mechanisms, including membrane disruption, oxidative stress, protein denaturation, and DNA damage, as shown in Figure 16.

However, the exact mechanism through which immobilized AgNPs kill bacteria is still not fully understood. However, it is thought that immobilized AgNPs may release silver ions (Ag+); this is likely one of the ways that AgNPs kill bacteria [71]. The silver ions (Ag+) are essential for the antibacterial and toxicity activities of silver (Ag). To maintain these activities, silver must be in its ionized state (Ag+) [72]. Ag+ can bind to nucleic acids, and they prefer to bind to nucleosides (the sugar-base units) rather than the phosphate groups [73]. This is why all silver-based materials that have antibacterial activity ultimately release silver ions [74]. Some studies have shown that positively charged nanoparticles (NPs) are attracted to negatively charged bacterial cells [75]. These NPs have been proposed to be very effective at killing bacteria [76]. Ag+ ions are attracted to sulfur-containing proteins in the cytoplasm and cell wall of bacteria. This attraction causes the Ag+ ions to attach to the bacteria and make their cell walls more permeable. This allows the Ag+ ions to enter the bacteria and kill them [77]. Once Ag+ enter bacteria, they deactivate the enzymes such as respiratory enzymes. This causes the bacteria to produce reactive oxygen species (ROS), which are toxic agents that damage the bacteria’s DNA and other important molecules. The Ag+ ions also prevent the bacteria from producing energy such as in interrupting adenosine triphosphate (ATP) release [78]. ROS can play a major role in damaging the cell membrane and deoxyribonucleic acid (DNA); the DNA is mainly made up of phosphorus and sulfur, and when it interacts with Ag+ ions, a number of difficulties can be caused, including the prevention of DNA replication and cell division. In addition, the Ag+ ions can also efficiently stop protein synthesis by denaturing cytoplasmic ribosomal components [79].

Furthermore, AgNPs can kill bacteria even if they are not Ag+ ions, which can damage the cell membranes of bacteria, and due to the nanoscale size, they can also enter bacteria and change the way the cell membrane works [79]. Therefore, immobilized AgNPs can kill bacteria by damaging their cell membranes and disrupting their signal transduction pathways, causing the organelles inside bacteria to rupture, which can lead to cell death. AgNPs can also dephosphorylate tyrosine residues on protein substrates, which can interfere with the bacteria’s signal transduction pathways. This disruption of signal transduction can lead to cell apoptosis (programmed cell death) and the inhibition of cell division [80].

## 4. Materials and Methods

### 4.1. Collection of Plant Materials

The gum of *M. oleifera* was gathered from healthy and fresh plants from the Medicinal Botanic Garden (MBG), Medicinal Botanic Center (MBC), Pakistan Council of Scientific and Industrial Research (PCSIR), Laboratories Complex, Peshawar, Pakistan. Furthermore, all methods were carried out according to institutional and national guidelines, which comply with international standards.

### 4.2. Authentication of Plant Materials

Samples were collected from three-year-old *M. oleifera* trees in March 2024 (Spring season), grown in clay soil with a suitable soil pH (5.8) and temperature range, from 28 to 32 ± 2 °C (a Moring tree picture is in the Appendix A). These plant materials (gum and *M. oleifera*) were recognized and verified by Dr. Hina Fazal (a specialist in plant taxonomy), and the herbarium specimen with Voucher No. MBC-PES-10811 was deposited in the herbarium of the Medicinal Botanic Center (MBC), Pakistan Council of Scientific and Industrial Research (PCSIR), Laboratories Complex, Peshawar, Pakistan.

### 4.3. Extract Preparation

For antimicrobial activity, active parts (gum) of the plants were powdered; 1.0, 2.5, and 5.0 g of powdered gum was dissolved in HPLC-grade water (Sigma; Aldrich; Darmstadt, Germany) to obtain a final volume of 10 mL. The colloidal solution was stored at 4 °C in airtight small bottles after accurate weighing before activities and its immobilization with silver nanoparticles.

### 4.4. Biogenic Synthesis of Silver Nanoparticles

Leaves of *M. oleifera* plant were collected from the Medicinal Botanic Garden (MBG), PCSIR Labs Complex, Peshawar, Pakistan. The leaves were oven-dried at 50 °C for 24 h and subsequently grinded with the help of a grinder. Distilled water (1000 mL) and powdered leaves (5 g) was boiled for 5 min, filtered twice, and then cooled at 4 °C for further usage. A silver nitrate (AgNO_3_) solution of 8.0 mM was prepared upon dissolving the appropriate amount of silver nitrate in deionized water. The solution was mixed using a vortex machine, and further dilutions of 6.0 mM, 4.0 mM, and 2.00 mM were made from the stock solution, following the equation C_1_V_1_ = C_2_V_2_. Plant extracts were combined with silver nitrate solutions in order to produce silver nanoparticles using the protocol in [81]. In a series of reactions including equal volumes of a twofold dilution of plant extract (8.0 mg/mL to 2.0 mg/mL) and AgNO_3_ (8.0 mg/mL to 2.0 mg/mL), various concentrations of plant extract and AgNO_3_ were mixed to select the most efficient concentration for the synthesis of AgNPs. Here, 5 mL of reactant in 15 mL tube was kept at room temperature for 24 h, and any change in color indicated the development of AgNPs. The best biosynthesized nanoparticle was selected based on the surface plasmon resonance and peak area. The biogenic nanoparticles were collected as pellets after centrifugation at 13,000 rpm for 15 min at room temperature. The supernatant was discarded, and the pellet was dissolved in deionized water and centrifuged again using the same protocol. This process was repeated thrice. The final supernatant was discarded, and the pellets were dried and stored in a refrigerator as biogenic nanoparticles.

### 4.5. Characterization of Silver Nanoparticles

#### 4.5.1. UV–Spectrophotometry

A UV–visible spectrophotometer was used to measure the formation of silver nanoparticles using light absorption between 300 and 600 nanometers [82].

#### 4.5.2. Fourier Transform Infrared Spectroscopy (FTIR)

FTIR analysis was performed to investigate the functional groups on the nanoparticles according to the protocol in [83]. Free biomass residues and non-capping ligand impurities were eliminated. The remaining solution was centrifuged at 10,000 rpm for 10 min, and the final suspension was reconstituted in 2 mL of distilled water. This centrifugation and redispersion process were repeated three times, after which the purified suspension was freeze-dried. The resulting powder was then used for FTIR analysis.

#### 4.5.3. X-Ray Diffraction

X-ray diffraction (XRD) analysis was employed to study the overall oxidation state and crystalline structure of silver nanoparticles [84]. Through the centrifugation and redispersion of the pellets into deionized water, prepared nanoparticles were purified and freeze-dried and analyzed through XRD.

#### 4.5.4. TEM Analysis

Transmission Electron Microscopy (TEM) was performed to investigate the size, morphology, and distribution of the silver nanoparticles. A drop of a sample was loaded on copper grids coated with carbon and placed on the grid. Further, the film on the grid was left to dry at room temperature and analyzed through a Transmission Electron Microscope. An elemental analysis of the biosynthesized silver nanoparticles conducted calibrated using an energy-dispersive X-ray (EDX) detector [83].

### 4.6. Selection of Nanoparticles

The silver nanoparticles (AgNPs) were selected for the growth inhibition of resistant microorganisms including both Gram-positive and Gram-negative; however, some reports suggested that these AgNPs can also inhibit the growth of fungi and viruses by disrupting the cell membrane and generating ROS [85].

### 4.7. Selection of Pathogens

In this study, M. oleifera gums and AgNPs were applied against *Bacillus subtilis* (*B. subtilis*), *Escherichia coli* (*E. coli*), *Klebsiella pneumoniae* (*K. pneumoniae*), *Proteus mirabilis* (*P. mirabilis*) *Pseudomonas aeruginosa* (*P. aeruginosa*), and *Salmonella typhi* (*S. typhi*), which cause various infections in humans [86,87,88,89,90]. These infectious microbes (Table 2) were received from different hospitals through the Pakistan Council of Scientific and Industrial Research (PCSIR) Laboratories Complex, Peshawar, Pakistan. These microorganisms were kept at 4 °C before antimicrobial activities.

### 4.8. Nutrient Agar Preparation

To make nutrient agar media, 900 mL of distilled water and 28 g of nutrient agar (15 g/L agar, 5 g/L gelatin component, 5 g/L NaCl, 1 g/L beef extract, and 2 g/L yeast extract) (Sigma, Aldrich, Darmstadt, Germany) were added to a 1 L screw-capped container. The solution was heated while being thoroughly agitated, the volume was increased to 1 L, and the bottle was sealed. The nutritional agar media bottle and the tools (Wire loops, Flasks, Glass tubes, Eppendorf tubes, swabs, cell spreaders, micropipette tips, and disks) that were used in the next tests were autoclaved at 121 °C for 20 min at 1.5 pounds per square inch (PSI) to sanitize them. The nutritional agar medium and sterilized equipment were added to the laminar flow unit (LFU) once it had cooled to 55 °C. Under sterile circumstances, the medium was transferred onto Petri dishes with approximately 35 milliliters of media applied to each plate. After being left for half an hour to form, the Petri dishes were covered with lids. The Petri dishes were turned upside down and incubated for 24 h at 37 °C to check for contamination. After that, these plates were put through testing for antibacterial activity [91]. Gram staining was performed for all isolated colonies according to the standard procedure in [92].

### 4.9. Antibiotic Susceptibility Test

The disk diffusion method and the Kirby–Bauer method of antibiotics was performed for the microbial sensitivity. Nutrient broth (1.3 g/100 mL) and nutrient agar (2.8 g/100 mL) were combined with hot distilled water to prepare the cultures for testing. The mixture was then transferred into test tubes (6–7 mL), Petri dishes (20 mL), and flasks (18–23 mL). To check for undesirable microbial development, the medium was sterilized at 121 °C for 20 min at 15 pressure and then incubated overnight at 37 °C. The microorganisms were streaked onto solid medium plates from the stock cultures, and cultured containers were placed for 24 h at 37 °C.

After being moved to liquid medium on the second day, the bacterial cultures were incubated for 10 h at 100 rpm in a GLSC-SBR-04-28 water bath at 32 °C. The optical density (OD) of the growing culture was maintained between 0.1 and 0.2, and the turbidity of the microbial growth in the test tubes was standardized by comparing it to the 0.5 McFarland standard. After that, solid media plates were covered with 0.1 mL of the standardized cultures, and they were refrigerated for 15 min to allow for absorption. Each cultivated plate’s surface was meticulously coated in three copies of a standard disk containing each antibiotic. A negative control disk was utilized, which lacked antibiotics. In order to determine if the bacterium is sensitive or resistant to antibiotics, the cultured plates were incubated for 24 h at 37 °C. The antimicrobial efficiency was then assessed using the millimeter diameter of the zones of inhibition surrounding the disks [93].

### 4.10. Antimicrobial Assay of Immobilized AgNPs

The physical mixing method was applied for the immobilization of AgNPs in gum of *M. oleifera*. In preliminary experiments, 10% to 80% gum solution was less effective for the encapsulation of AgNPs. Here, 85% was the best candidate for the immobilization process, while more than 85% was unable to move/flow in suspension. The different concentrations of AgNPs suspension were maintained using a magnetic stirrer/ultrasonicator, and 85% gum solution (slightly heated) was added to the suspension drop-wise, followed by cooling to form round shaped bodies for easy attachment to the bacterial cell. After the successful uniform and homogenous polymerization of NPs with gum, differential AgNPs were allowed for immobilization with gum (85%).

In this study, six different bacterial strains were used for antimicrobial assay. These bacterial strains included *B. subtilis*, *E coli*, *K pneumoniae*, *P. mirabilis*, *P. aeruginosa*, and *S. typhi*. The well diffusion method was utilized for the antibacterial activity of nanoparticles and gums, and the procedure was repeated for synergistic combinations/differential ratios of nanoparticles + gums (Table 3) [94]. The zone of inhibition was measured in mm as the diameter around the well. The bacterial strains were streaked in plates, and the wells were bored in each plate.

The following treatments contained immobilized AgNPs with gum of *M. oleifera*: T1 (25 µL AgNPs + 200 µL gums), T2 (50 µL AgNPs + 200 µL gums), T3 (75 µL AgNPs + 200 µL gums), T4 (100 µL AgNPs + 200 µL gums), T5 (125 µL + 200 µL gums), T6 (150 µL AgNPs + 200 µL gums), T7 (175 µL AgNPs + 200 µL gums), T8 (200 µL AgNPs + 200 µL gums), T9 (225 µL AgNPs + 200 µL gums), T10 (250 µL AgNPs + 200 µL gums), C1 (100 µL AgNPs alone), and C2 (200 µL gums alone). These combinations were applied for the determination of antimicrobial activities. All the plates (containing bacteria and a combination of AgNPs + gum) were kept at 37 °C in an incubator for one day. After that, the zones of inhibition were recorded in millimeters (mm).

### 4.11. Statistical Analysis

The antimicrobial activity of different treatments was measured three times. Excel was used to calculate descriptive statistics (mean values ± standard error). One-way analysis of variance (ANOVA) was conducted through Statistics software (Version 8.1; Tallahassee, FL, USA) (V. 8.1; USA) and for determining significance and least significant differences. Origin Lab software (Version 8.5; Northampton, MA, USA) was used to create graphical representations of the data.

## 5. Conclusions

The immobilized AgNPs displayed promising potential by inhibiting the growth of highly resistant *B. subtilis*, *E. coli*, *K. pneumoniae*, *P. mirabilis*, *P. aeruginosa*, and *S. typhi*. These results conclude that the immobilized AgNPs have proven antimicrobial activities; however, the gum of *M. oleifera* may serve as a stabilizing or immobilizing agent for silver nanoparticles, enhancing their antibacterial effects. Silver nanoparticles are known to exhibit antimicrobial properties by interfering with bacterial cell membranes and other cellular structures, leading to the inhibition of bacterial growth. The immobilization strategy enhanced the antimicrobial efficacy that may be attributed to the natural polymerization of gum and their stability and controlled release of silver ions. Further research and validation are needed to confirm the effectiveness, understand the underlying mechanisms, and assess the safety of such formulations for practical applications, especially in medical or environmental contexts. This study shows the nanoparticles’ potential to be adopted by pharmaceutical industries as highly effective antibacterial alternatives to synthetic drugs.

## Figures and Tables

**Figure 1 pharmaceuticals-17-01546-f001:**
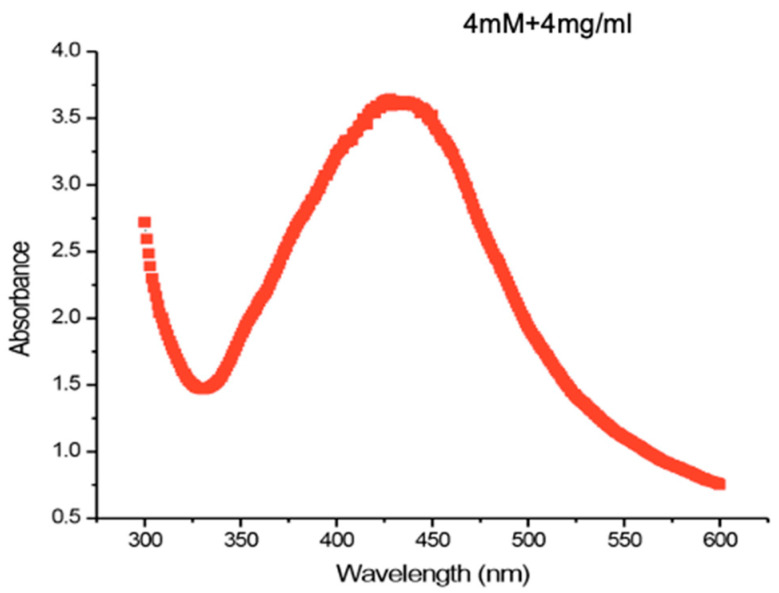
UV–visible spectroscopy of Moringa leaf-derived silver nanoparticles.

**Figure 2 pharmaceuticals-17-01546-f002:**
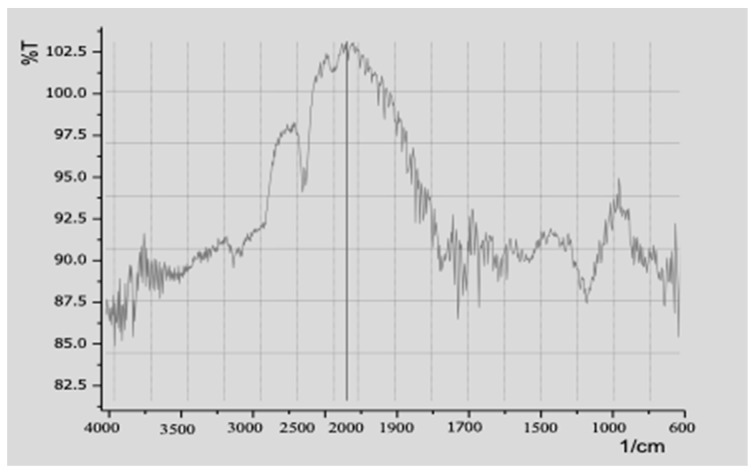
FTIR spectra of Moringa leaf (ML)-derived silver nanoparticles.

**Figure 3 pharmaceuticals-17-01546-f003:**
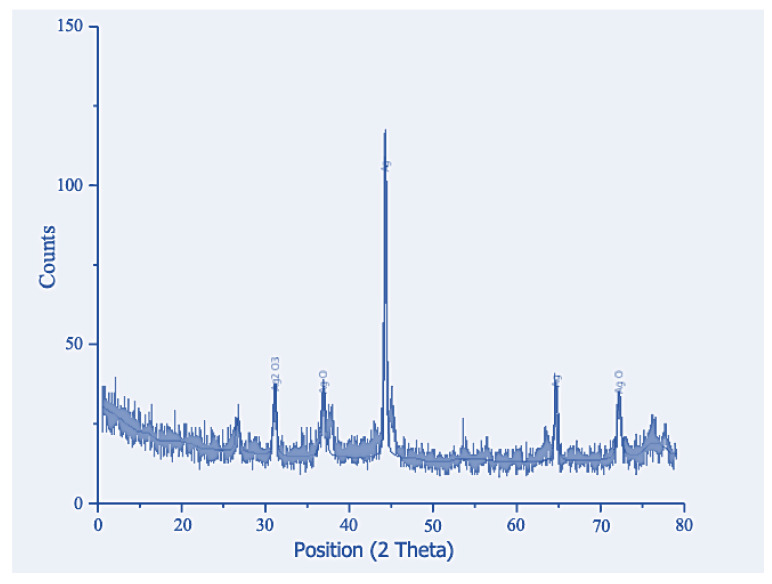
XRD spectra of ML-derived silver nanoparticles.

**Figure 4 pharmaceuticals-17-01546-f004:**
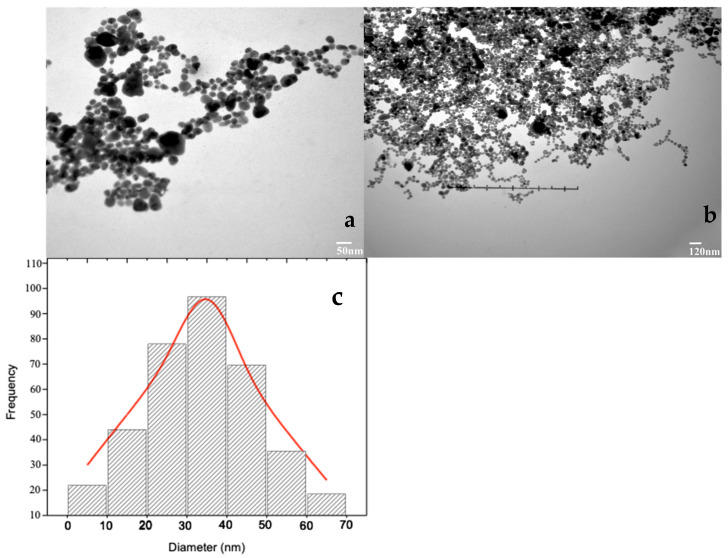
TEM images of ML-derived silver nanoparticles: (**a**) 50 nm, (**b**) 120 nm, and (**c**) histogram.

**Figure 5 pharmaceuticals-17-01546-f005:**
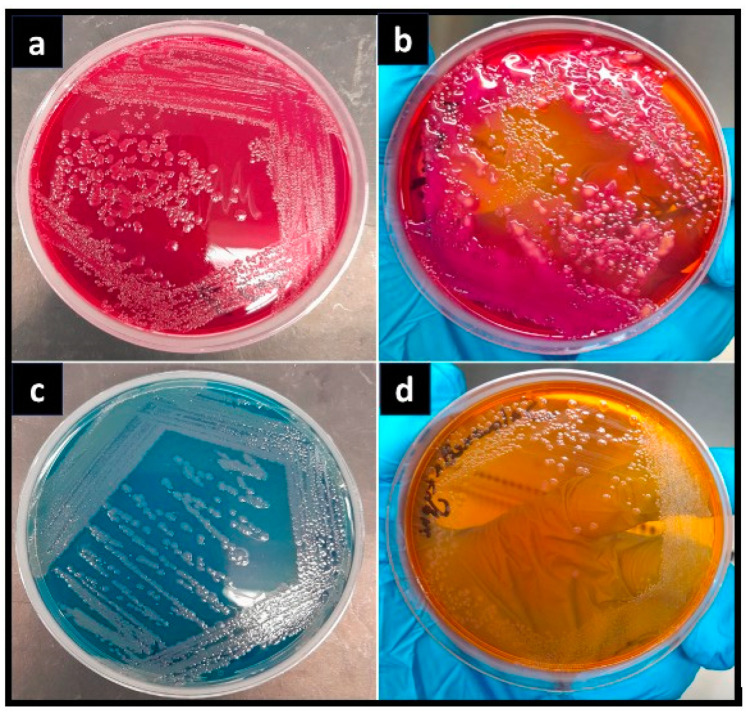
Morphological representation of different bacterial strains on MAC agar and CLED agar plates. (**a**) *E. coli*, (**b**) *K. pneumoniae* (**c**) *P. aeruginosa*, and (**d**) *S. typhi*.

**Figure 6 pharmaceuticals-17-01546-f006:**
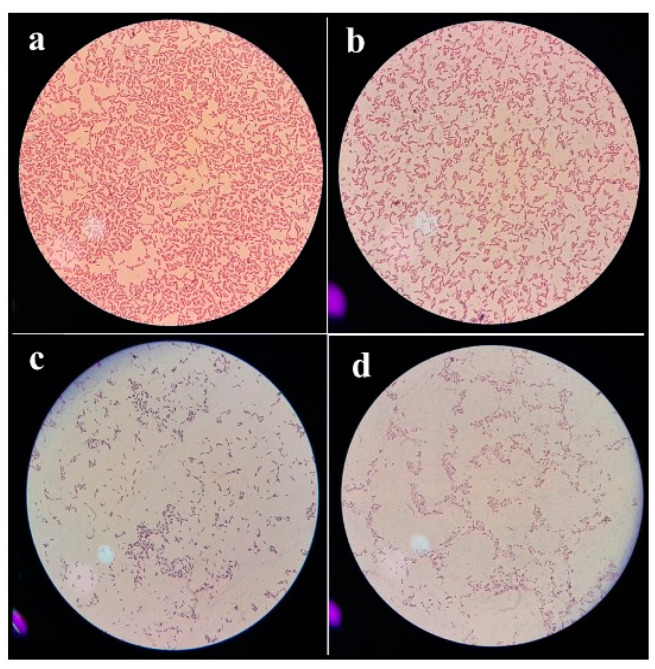
Microscopic representation of different bacterial strains. (**a**) *E. coli*, (**b**) *K. pneumoniae*, (**c**) *P. aeruginosa*, and (**d**) *S. typhi*.

**Figure 7 pharmaceuticals-17-01546-f007:**
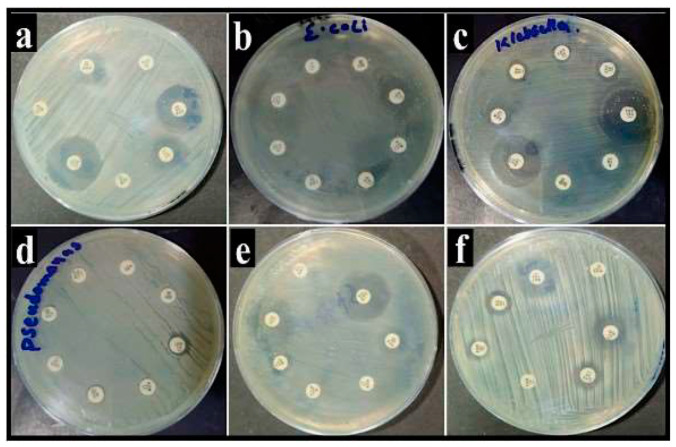
Pictorial representation of zones of inhibition of various drugs against different bacterial strains (**a**) *B. subtilis*, (**b**) *E. coli*, (**c**) *K. pneumoniae*, (**d**) *P. aeruginosa*, (**e**) *P. mirabilis*, and (**f**) *S. typhi*.

**Figure 8 pharmaceuticals-17-01546-f008:**
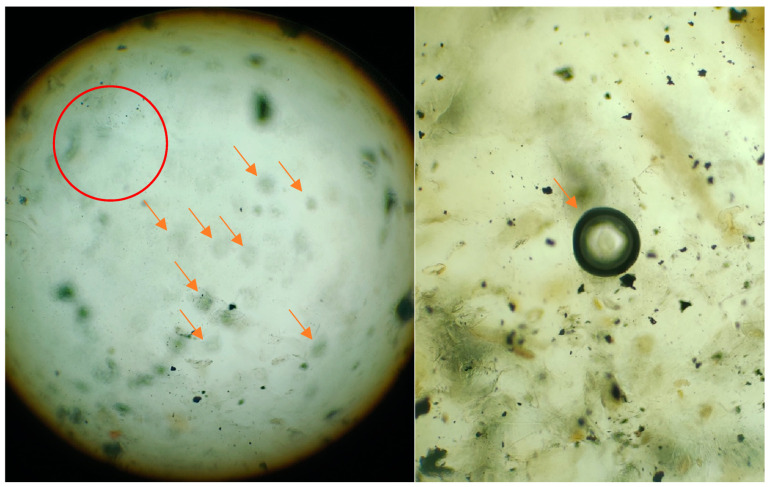
Micrographs of immobilization of silver nanoparticles and gum of *M. oleifera*. The red circle indicates the encapsulation of NPs in gum, while red arrows refers to strong binding of gum with maximum AgNPs, and the black circle (indicated by red arrow) represent the accumulation of NPs that appear in circular shape after zooming.

**Figure 9 pharmaceuticals-17-01546-f009:**
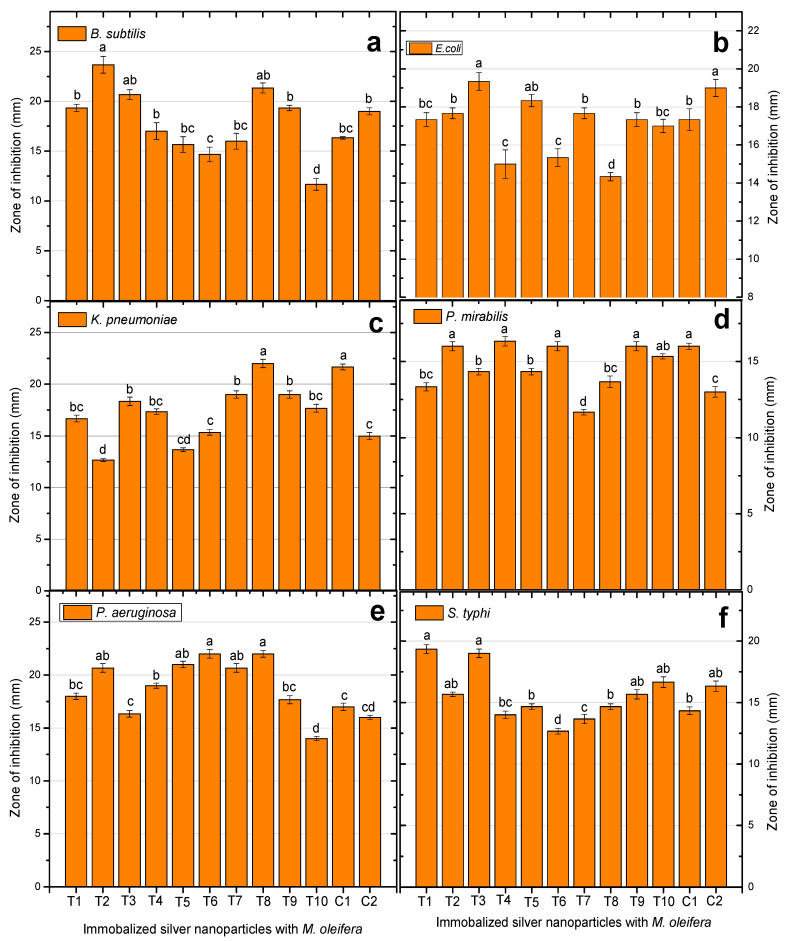
Antibacterial potential of different treatments of immobilized AgNPs (T1: 25 µL AgNPs + 200 µL gums; T2: 50 µL AgNPs + 200 µL gums; T3: 75 µL AgNPs + 200 µL gums; T4: 100 µL AgNPs + 200 µL gums; T5: 125 µL + 200 µL gums; T6: 150 µL AgNPs + 200 µL gums; T7: 175 µL AgNPs + 200 µL gums; T8: 200 µL AgNPs + 200 µL gums; T9: 225 µL AgNPs + 200 µL gums; T10: 250 µL AgNPs + 200 µL gums; AgNPs alone and gums alone against (**a**) *B. subtilis*, (**b**) *E. coli*, (**c**) *K. pneumoniae*, (**d**) *P. mirabilis*, (**e**) *P. aeruginosa*, and (**f**) *S. typhi*). Bars with alphabets and standard errors represents least significant differences among mean values from triplicates.

**Figure 10 pharmaceuticals-17-01546-f010:**
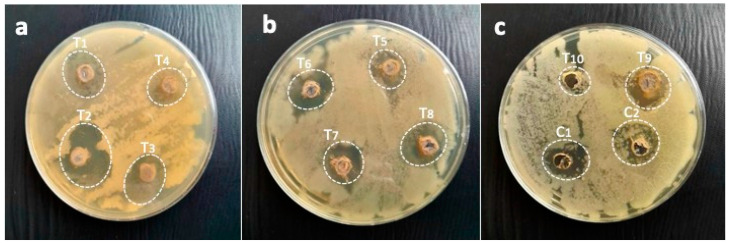
Pictorial presentation of zones of inhibitions of immobilized AgNPs against *B. subtilis* as (**a**) T1 (25 µL AgNPs + 200 µL gums), T2 (50 µL AgNPs + 200 µL gums), T3 (75 µL AgNPs + 200 µL gums), and T4 (100 µL AgNPs + 200 µL gums); (**b**) T5 (125 µL + 200 µL gums), T6 (150 µL AgNPs + 200 µL gums), T7 (175 µL AgNPs + 200 µL gums), and T8 (200 µL AgNPs + 200 µL gums); and (**c**) T9 (225 µL AgNPs + 200 µL gums), T10 (250 µL AgNPs + 200 µL gums), C1 (100 µL AgNPs alone), and C2 (200 µL gums alone).

**Figure 11 pharmaceuticals-17-01546-f011:**
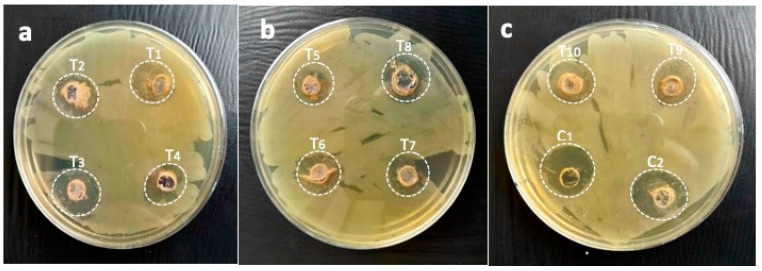
Pictorial presentation of zones of inhibitions of immobilized AgNPs against *E. coli* as (**a**) T1 (25 µL AgNPs + 200 µL gums), T2 (50 µL AgNPs + 200 µL gums), T3 (75 µL AgNPs + 200 µL gums), and T4 (100 µL AgNPs + 200 µL gums); (**b**) T5 (125 µL + 200 µL gums), T6 (150 µL AgNPs + 200 µL gums), T7 (175 µL AgNPs + 200 µL gums), and T8 (200 µL AgNPs + 200 µL gums); and (**c**) T9 (225 µL AgNPs + 200 µL gums), T10 (250 µL AgNPs + 200 µL gums), C1 (100 µL AgNPs alone), and C2 (200 µL gums alone).

**Figure 12 pharmaceuticals-17-01546-f012:**
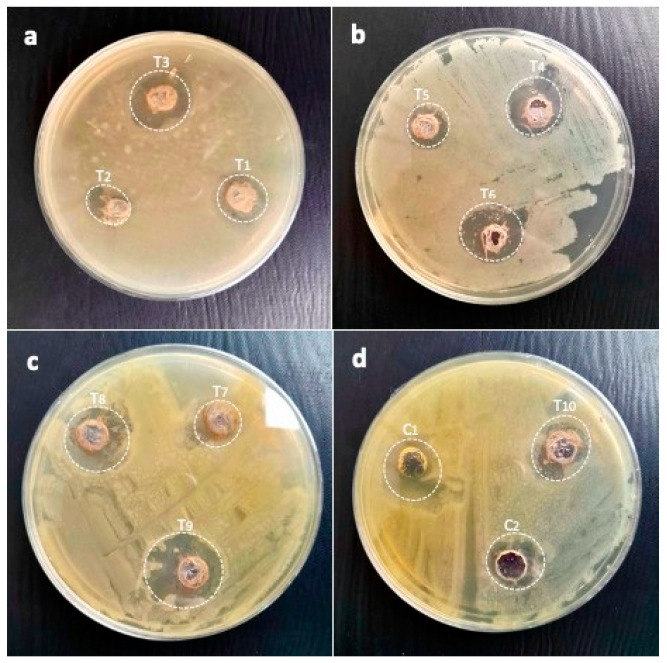
Pictorial presentation of zones of inhibitions of immobilized AgNPs against *K. pneumoniae* as (**a**) T1 (25 µL AgNPs + 200 µL gums), T2 (50 µL AgNPs + 200 µL gums), and T3 (75 µL AgNPs + 200 µL gums); (**b**) T4 (100 µL AgNPs + 200 µL gums), T5 (125 µL + 200 µL gums), and T6 (150 µL AgNPs + 200 µL gums); (**c**) T7 (175 µL AgNPs + 200 µL gums), T8 (200 µL AgNPs + 200 µL gums), and T9 (225 µL AgNPs + 200 µL gums); and (**d**) T10 (250 µL AgNPs + 200 µL gums), C1 (100 µL AgNPs alone), and C2 (200 µL gums alone).

**Figure 13 pharmaceuticals-17-01546-f013:**
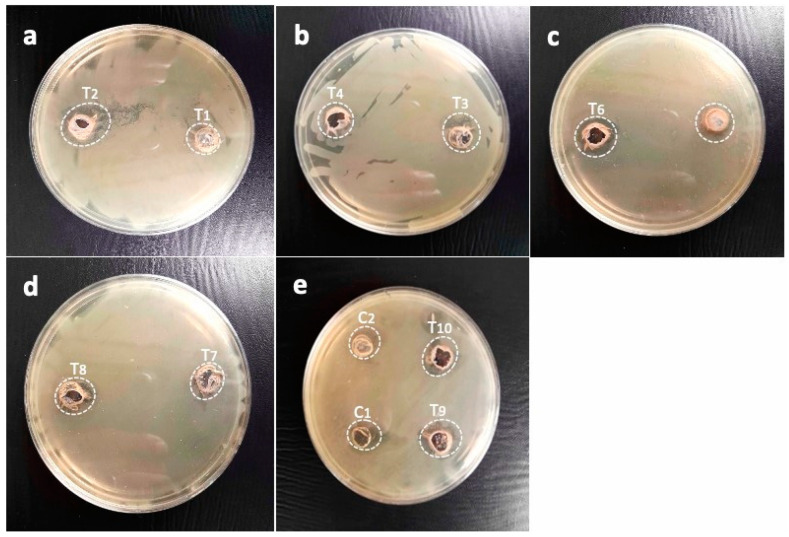
Pictorial presentation of zones of inhibitions of immobilized AgNPs against *P. mirabilis* as (**a**) T1 (25 µL AgNPs + 200 µL gums) and T2 (50 µL AgNPs + 200 µL gums); (**b**) T3 (75 µL AgNPs + 200 µL gums) and T4 (100 µL AgNPs + 200 µL gums); (**c**) T5 (125 µL + 200 µL gums) and T6 (150 µL AgNPs + 200 µL gums); (**d**) T7 (175 µL AgNPs + 200 µL gums) and T8 (200 µL AgNPs + 200 µL gums; and (**e**) T9 (225 µL AgNPs + 200 µL gums), T10 (250 µL AgNPs + 200 µL gums), C1 (100 µL AgNPs alone), and C2 (200 µL gums alone).

**Figure 14 pharmaceuticals-17-01546-f014:**
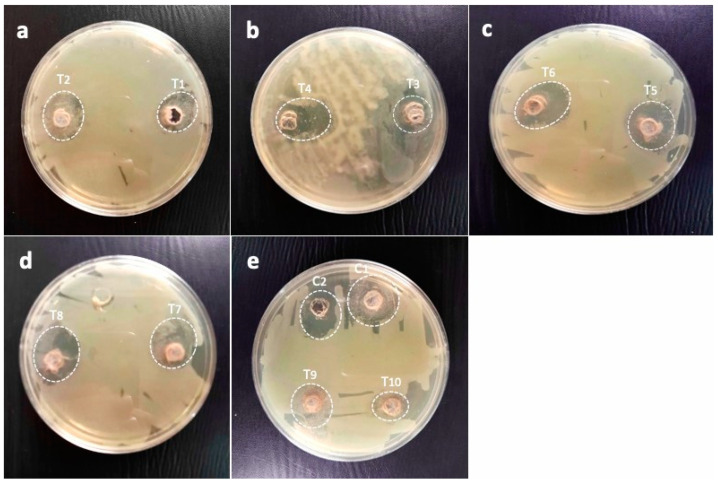
Pictorial presentation of zones of inhibitions of immobilized AgNPs against *P. aeruginosa* as (**a**) T1 (25 µL AgNPs + 200 µL gums) and T2 (50 µL AgNPs + 200 µL gums); (**b**) T3 (75 µL AgNPs + 200 µL gums) and T4 (100 µL AgNPs + 200 µL gums); (**c**) T5 (125 µL + 200 µL gums) and T6 (150 µL AgNPs + 200 µL gums); (**d**) T7 (175 µL AgNPs + 200 µL gums) and T8 (200 µL AgNPs + 200 µL gums); and (**e**) T9 (225 µL AgNPs + 200 µL gums), T10 (250 µL AgNPs + 200 µL gums), C1 (100 µL AgNPs alone), and C2 (200 µL gums alone).

**Figure 15 pharmaceuticals-17-01546-f015:**
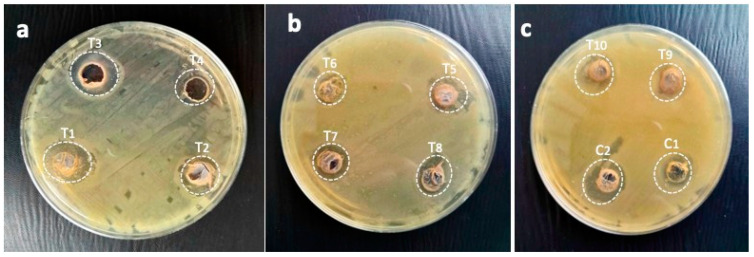
Pictorial presentation of zones of inhibitions of immobilized AgNPs against *S. typhi* as (**a**) T1 (25 µL AgNPs + 200 µL gums), T2 (50 µL AgNPs + 200 µL gums), T3 (75 µL AgNPs + 200 µL gums), and T4 (100 µL AgNPs + 200 µL gums); (**b**) T5 (125 µL + 200 µL gums), T6 (150 µL AgNPs + 200 µL gums), T7 (175 µL AgNPs + 200 µL gums), and T8 (200 µL AgNPs + 200 µL gums); and (**c**) T9 (225 µL AgNPs + 200 µL gums), T10 (250 µL AgNPs + 200 µL gums), C1 (100 µL AgNPs alone), and C2 (200 µL gums alone).

**Figure 16 pharmaceuticals-17-01546-f016:**
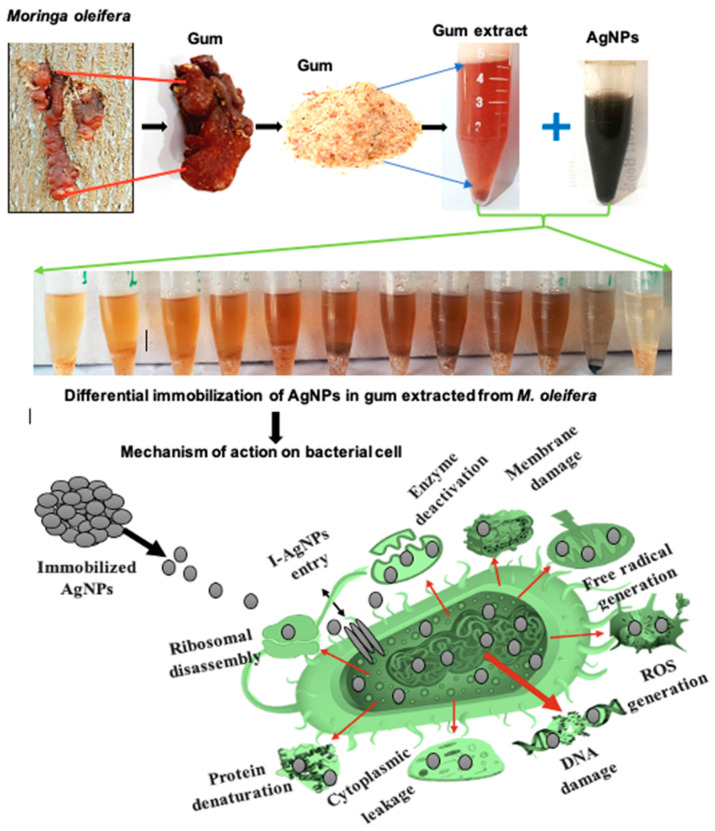
Mechanism of bacterial cell death through the application of immobilized AgNPs.

**Table 1 pharmaceuticals-17-01546-t001:** Antibiotic sensitivity status and zone of inhibition in mm of different drugs.

Pathogenic Species			Zones of Inhibition (mm) of Different Antibiotics					
	Do	CRO	CTX	SCF	E	SXT	IPM	AZM
*B. subtilis*	10.1 ± 0.1	12.2 ± 0.4	15.3 ± 0.3	17.3 ± 0.3	**R**	**R**	---	**R**
*E. coli*	8.3 ± 0.1	10.2 ± 0.1	13.5 ± 0.4	**R**	**R**	14.6 ± 0.5	15.1 ± 0.3	**R**
*K. pneumoniae*	10.5 ± 0.2	**R**	10.6 ± 0.6	17.3 ± 0.3	**R**	**R**	15.4 ± 0.7	10.2 ± 0.4
*P. aeruginosa*	8.9 ± 0.2	**R**	**R**	**R**	**R**	**R**	10.2 ± 0.8	**R**
*P. mirabilis*	18.2 ± 0.4	**R**	10.3 ± 0.1	18.4 ± 0.6	---	**R**	12.4 ± 0.2	**R**
*S. typhi*	14.3 ± 0.1	**R**	10.2 ± 0.3	15.2 ± 0.5	---	**R**	12.3 ± 0.5	**R**

**Table 2 pharmaceuticals-17-01546-t002:** Collected bacterial strains, voucher specimen numbers, strain types, and sources. The strains were Gram-positive (G+ve) and Gram-negative (G-ve).

Voucher Specimen Numbers	Bacterial Strain	Types	Source
MBC-MIC-208	*B. subtilis*	G + ve	PCSIR Peshawar
MBC-MIC-003	*E. coli*	G − ve	PCSIR Peshawar
MBC-MIC-459	*K. pneumoniae*	G − ve	PCSIR Peshawar
MBC-MIC-405	*P. mirabilis*	G − ve	PCSIR Peshawar
MBC-MIC-051	*P. aeruginosa*	G − ve	PCSIR Peshawar
MBC-MIC-104	*S. typhi*	G − ve	PCSIR Peshawar

**Table 3 pharmaceuticals-17-01546-t003:** Differential treatments/concentrations of AgNPs and gum of *M. oleifera* for immobilization.

Treatments	Applied Combinations
T1	25 µL AgNPs + 200 µL gums
T2	50 µL AgNPs + 200 µL gums
T3	75 µL AgNPs + 200 µL gums
T4	100 µL AgNPs + 200 µL gums
T5	125 µL AgNPs + 200 µL gums
T6	150 µL AgNPs + 200 µL gums
T7	175 µL AgNPs + 200 µL gums
T8	200 µL AgNPs + 200 µL gums
T9	225 µL AgNPs + 200 µL gums
T10	250 µL AgNPs + 200 µL gums
C1	100 µL AgNPs alone
C2	200 µL gums alone

## Data Availability

The presented data in the current manuscript will be provided upon request to the corresponding author.

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
