# Peer review of "Immobilization of Silver Nanoparticles with Defensive Gum of Moringa oleifera for Antibacterial Efficacy Against Resistant Bacterial Species from Human Infections"

_pharmaceuticals, 2024, doi:10.3390/ph17111546_

Round 1
Reviewer 1 Report
Comments and Suggestions for Authors
The submitted manuscript “Immobilization of silver nanoparticles with gum of Moringa oleifera for fast and extremely effective antibacterial efficacy against highly resistant bacterial species causing various infections in human” focus on an interesting and relevant subject regarding human health and antimicrobials resistance. Considering nowadays pertinence of the subject in the “one health” perspective, is important to find new strategies against antimicrobial resistance and analyse scientific data that can be used by pharmaceutical industry.
In general, the conceptualization of the study is correct. However, the small number of bacterial strains elected to the analysis compromises the relevance of the present study and should be considered as a main limitation.
Title – I suggest to correct the title accordingly to the limitations: “Immobilization of silver nanoparticles with gum of Moringa oleifera for antibacterial efficacy against resistant bacterial species from human infections”
The Abstract must be improved.
The authors should provide information regarding methodology.
The abstract should provide correct information regarding the content of the manuscript (objectives, materials and methods, results and discussion) in a clear and simple language.
The keywords are not adequate to the content of the study.
“Misuse of antibiotics; antibiotic resistance” are not correct to this study.
The introduction but can be improved.
The information about Moringa oleifera should be reduced: specific and related to antimicrobial properties.
The action mechanism of AgNPs against bacteria should be developed in introduction section.
The results regarding Biogenic synthesis of nanoparticles and its characterization are correctly exposed.
The bacterial isolates should be identified by more accurate and reliable methods (API (analytical profile index), Vitek system or similar).
Even tough the bacteria isolates used were considered multiresistant (MDR) being resistant to 3 or more classes of antimicrobials, the number of antimicrobials tested are considered insufficient.
The Discussion section is well structured and explains the results of the study.
The methodology should be improved, by increasing the number of bacterial strains tested. Also increasing the number of antimicrobials tested in each strain. Additionally, the authors do not explain which tables of resistance were used to consider resistance (CLSI; EUCAST?).
The authors refer the negative control, using a blank disk. However, a positive control (a disk with proven antimicrobial sensibility for each isolate should also be included in each plate).
In my opinion, the preparation of culture media (point 4.8) in not necessary. The Gram staining procedure is not necessary to be described (4.10-4.12). The authors just have to mention that a Gram staining was performed as described by….
The information regarding the infections caused by each type of bacteria is not necessary. “B. subtilis caused lung infection, septicemia endocarditis, pneumonia, bacteremia, and in immune compromised individuals [86]. E. coli cause urinary tract infection (UTI) and intestinal infection (vomiting, abdominal cramps, and bloody diarrhea) [87]. K. pneumoniae causes nosocomial infection of blood stream, respiratory system infection, UTI, and premature child’s intensive care unit disease/infections [88]. Urinary tract infections linked with catheter usage and bloodstream infections (CAUTIs) are caused by P. mirabilis [89]. P. aeruginosa can result in infections that develop in the blood, lungs (pneumonia), or other parts of the body [86]. Typhoid fever is caused by S. typhi, an intestinal infection [90].” I suggest to remove the above sentences.
The conclusions are sound and cautious considering the result of the study.
Author Response
Response letter to Editor and Reviewers comments
Dear Dr. Ahmad,
Thank you again for your manuscript submission:
Manuscript ID: pharmaceuticals-3228584
Type of manuscript: Article
Title: Immobilization of silver nanoparticles with gum of Moringa oleifera
for fast and extremely effective antibacterial efficacy against highly
resistant bacterial species causing various infections in human
Please revise the manuscript found at the above link according to the
reviewers' comments and upload the revised file within 10 days.
Kind regards,
Ms. Celine Wan
E-Mail: celine.wan@mdpi.com
Reviewer 1
The submitted manuscript “Immobilization of silver nanoparticles with gum of Moringa oleifera for fast and extremely effective antibacterial efficacy against highly resistant bacterial species causing various infections in human” focus on an interesting and relevant subject regarding human health and antimicrobials resistance. Considering nowadays pertinence of the subject in the “one health” perspective, is important to find new strategies against antimicrobial resistance and analyse scientific data that can be used by pharmaceutical industry.
Dear Professor,
Thank you very much for your valuable comments. Your comments really impressed me. We are happy for the chance to eliminate these weaknesses in our manuscript. Dear Sir, we have tried our best to incorporate all of changes/modifications suggested by you.
Comment: In general, the conceptualization of the study is correct. However, the small number of bacterial strains elected to the analysis compromises the relevance of the present study and should be considered as a main limitation.
Response: We only selected the resistant strains for the study rather than using a huge number of microorganisms. One of the reviewers and editor commented that why several strains were selected. We received contrasting comments. As compared to other studies, we presented the activities against the maximum number of resistant strains. We think that the current MS will be a model article for future studies.
Comment: Title – I suggest to correct the title accordingly to the limitations: “Immobilization of silver nanoparticles with gum of Moringa oleifera for antibacterial efficacy against resistant bacterial species from human infections”
Response: Strongly agreed. We revised the title accordingly as “Immobilization of silver nanoparticles with defensive gum of Moringa oleifera for antibacterial efficacy against resistant bacterial species from human infections”
Comment: The Abstract must be improved.
Response: The abstract has been revised accordingly.
Comment: The authors should provide information regarding methodology.
Response: corrected as suggested
Comment: The abstract should provide correct information regarding the content of the manuscript (objectives, materials and methods, results and discussion) in a clear and simple language.
Response: We have revised the abstract accordingly for easy reading and for more clarity.
Comment: The keywords are not adequate to the content of the study.
Response: Thanks for nice suggestions. The key words have been revised as “Moringa oleifera, organic synthesis of silver nanoparticles, defensive gum, immobilization of AgNPs, Antibiotic resistance, antibacterial activity, alternative antimicrobials.”
Comment: “Misuse of antibiotics; antibiotic resistance” are not correct to this study.
Response: We have revised and removed these words accordingly.
Comment: The introduction but can be improved.
Response: Revised as suggested
Comment: The information about Moringa oleifera should be reduced: specific and related to antimicrobial properties.
Response: We have reduced the irrelevant information’s from introduction as suggested.
Comment: The action mechanism of AgNPs against bacteria should be developed in introduction section.
Response: Strongly agreed. A short story of mechanism of action has been incorporated in introduction section as “The AgNPs alone have proven antimicrobial potential while the gum is released in response to pathogens attack in plants. Here, the dual and synergistic potential of AgNPs along with gum was investigated for the growth inhibition of infectious strains. The immobilized AgNPs have potential to kill infectious strains by disrupting the membranes by inducing pores and cause cytoplasmic leakage and bacterial cell death. It is possible that immobilized AgNPs can stimulate the defense system of bacterial cell more rapidly than either NPs alone or using gum alone. The immobilization strategy most probably release a huge quantities of reactive oxygen species or other radical that directly react and inhibits the major functions such metabolism, replication, translation, denaturation, and DNA damage.”
Comment: The results regarding Biogenic synthesis of nanoparticles and its characterization are correctly exposed.
Response: Thank you so much for your valuable encouragement.
Comment: The bacterial isolates should be identified by more accurate and reliable methods (API (analytical profile index), Vitek system or similar).
Response: Yes Sir, we also used API-10 S (BIOMERIEUX) kit (Made in France) with proper ONPG, GLU, ARA. LDC, ODC, CIT, H2S, URE, TDA, IND, OX and NO2. We used multiple methods for the identification because we are further working on the whole genome sequencing to identify the mutated gene that may be OXA, CTX, Beta-lactamase to overcome the escalating resistance and to develop alternative antimicrobials or to modify the existing antibiotics by changing its action site.
Comment: Even tough the bacteria isolates used were considered multiresistant (MDR) being resistant to 3 or more classes of antimicrobials, the number of antimicrobials tested are considered insufficient.
Response: Only the selected antibiotics were applied in this study after communication with different physicians and surgeons facing the rapidly growing infections. They already used multiple classes of antibiotics alone or in synergism (rare-sensitive with intermediate) and recently they received funds from WHO to investigate and to identify the most effective way to control these infections. That is why we only used selected antibiotics after their recommendation.
Comment: The Discussion section is well structured and explains the results of the study.
Response: Thank you Sir.
Comment: The methodology should be improved, by increasing the number of bacterial strains tested. Also increasing the number of antimicrobials tested in each strain. Additionally, the authors do not explain which tables of resistance were used to consider resistance (CLSI; EUCAST?).
Response: Confusing comment.
Comment: The authors refer the negative control, using a blank disk. However, a positive control (a disk with proven antimicrobial sensibility for each isolate should also be included in each plate).
Response: We already mentioned a double positive control for each bacterial strain.
Comment: In my opinion, the preparation of culture media (point 4.8) in not necessary. The Gram staining procedure is not necessary to be described (4.10-4.12). The authors just have to mention that a Gram staining was performed as described by….
Response: Strongly agreed. We have removed the following sections accordingly
4.8. Preparation of culture media
4.8.1. Blood Agar Preparation
To prepare media, 900 ml of distilled water and 20 g of blood agar (10.0 g/l meat extract, 10.0 g/l Tryptose, 5.0 g/l sodium chloride, and 15.0 g/l agar) purchased from (Zona lnd.le – Roseto d. Abruzzi (TE) - ITALY), were added to a 1-liter screw-capped bottle. The solution was heated while being thoroughly agitated, the volume was increased to 1 L, and the bottle was sealed and autoclaved at 121 ºC for 15 minutes. The blood agar medium was placed in the laminar flow unit (LFU) once it had cooled to 50 °C, 50 ml of sheep blood were added to the medium under control condition, the medium was transferred onto petri dishes. the petri dishes were covered with lids. The petri dishes were turned upside down and incubated for 24 hours at 37 °C to check for contamination.
4.8.2. Cystine Lactose Electrolyte Deficient Agar (CLED agar)
CLED- agar contains 0.128 gram of L-cystine, 10 grams of lactose, 4 grams of tryptone, 0.02 grams of bromothymol blue, and 4 grams of peptone per liter of solution from (Oxoid, Basingstoke, Hampshire, England). To prepare the agar, 36.2 grams of the mixture were dissolved in distilled water and sterilized by autoclaving at 121 ºC for 15 minutes.
4.8.3. Muller Hinton Agar (MHA) preparation
MHA contains (Typical g/L= Beef, dehydrated infusion from 300.0, Casein hydrolysate 17.5, Starch 1.5 and Agar 17.0) from Oxoid Ltd. Wade Road, Basingstoke, Hants, Rg24. 8PW. UK). For media preparation 38 g dehydrate medium was dissolved in one liter of distilled water and boil the mixture until completely dissolved. Then, autoclave it, at 121ºC for 15 minutes. Afterward, pour 15-20 ml of the media into a 90 ml Petri dish and place it on a level surface to ensure uniform depth. The dehydrated media should be stored at room temperature, while the prepared media should be checked for sterility by incubation overnight and stored at 2-8ºC.
The topics and subtopics have been revised accordingly.
We also removed the following as suggested in the methodology
4.10 Bacterial inoculation
The provided protocol was followed, and a calibrated wire loop of 1 µl (0.001 ml) was utilized for specimen inoculation on MAC gar and CLED agar. This agar plates were incubated at 37 ºC for 24 hours, and any changes in color were observed after the incubation period for identification.
4.11. Gram staining
Gram staining was performed for all isolated colonies according to the standard procedure by [92].
4.12. Preparation and fixation of smear
To prepare and fix the smear, a drop of normal saline was placed on a sterile glass slide, and an isolated colony was picked from the culture plate using a sterile wire loop and placed on the slide. The inoculum was spread in a thin smear of 25-30 mm from the center to the periphery and heated to fix it by passing over the flame.
4.13. Staining of the fixed smear
To stain the fixed smear, crystal violet solution was added to the smear for one minute and the washed with tap water. Gram’s iodine was added as a mordant, and the smear was decolorized with 95% ethyl alcohol and rinsed with water. Finally, safranin was used as a counter stain for 60-80 seconds, and the slide was washed with tap water. The smear was allowed to air dry and then observed under the Microscope.
Comment: The information regarding the infections caused by each type of bacteria is not necessary. “B. subtilis caused lung infection, septicemia endocarditis, pneumonia, bacteremia, and in immune compromised individuals [86]. E. coli cause urinary tract infection (UTI) and intestinal infection (vomiting, abdominal cramps, and bloody diarrhea) [87]. K. pneumoniae causes nosocomial infection of blood stream, respiratory system infection, UTI, and premature child’s intensive care unit disease/infections [88]. Urinary tract infections linked with catheter usage and bloodstream infections (CAUTIs) are caused by P. mirabilis [89]. P. aeruginosa can result in infections that develop in the blood, lungs (pneumonia), or other parts of the body [86]. Typhoid fever is caused by S. typhi, an intestinal infection [90].” I suggest to remove the above sentences.
Response: The irrelevant information has been removed and revised it as “In this study, M. oleifera gums and AgNPs was applied against Bacillus subtilis (B. subtilis), Escherichia coli (E. coli), Klebsiella pneumoniae (K. pneumoniae), Proteus mirabilis (P. mirabilis) Pseudomonas aeruginosa (P. aeruginosa), and Salmonella typhi (S. typhi) that causes various infections in human [86-90]. These infectious microbes (Tab 1.2) were received from different hospitals through Pakistan Council of Scientific and Industrial Research (PCSIR) Laboratories Complex, Peshawar, Pakistan. These microorganisms were kept at 4 °C before antimicrobial activities.”
Comment: The conclusions are sound and cautious considering the result of the study.
Response: We have revised the conclusion as “The immobilized AgNPs displayed promising potential by inhibiting the growth of highly resistant B. subtilis, E. coli, K. pneumoniae, P. mirabilis, P. aeruginosa and S. typhi. These results concluded that the immobilized AgNPs have proven antimicrobial activities, however, the gum of M. oleifera may serve as a stabilizing or immobilizing agent for silver nanoparticles, enhancing their antibacterial effects. Silver nanoparticles are known to exhibit antimicrobial properties by interfering with bacterial cell membranes and other cellular structures, leading to the inhibition of bacterial growth. The immobilization strategy enhanced the antimicrobial efficacy that may be attributed to natural polymerization of gum and their stability and controlled release of silver ions. Further research and validation are needed to confirm the effectiveness, understand the underlying mechanisms, and assess the safety of such formulations for practical applications, especially in medical or environmental contexts. The study has a potential to be adopted by pharmaceutical industries as highly effective antibacterial alternatives to synthetic drugs.”

Reviewer 2 Report
Comments and Suggestions for Authors
Dear Editor,
the manuscript “Immobilization of silver nanoparticles with gum of Moringa oleifera for fast and extremely effective antibacterial efficacy against highly resistant bacterial species causing various infections in human” reported the preparation of antibacterial immobilized Ag NPs on Moringa oleifera-extracted gums against different pathogenic strains.
The manuscript will be reviewed again after major revision.
My comments are:
1-The idea is not completely novel. I see many Ag NPs immobilized on this gum, please define the novelty by the end of the introduction part and compare your results with other reported ones
2- The title is too long, please concise it
3-Abstract should be revised again and rewritten concisely
4-E. coli (19.3; 75 µL:200) in the abstract should be changed to E. coli (19.3 mm; 75 µL:200), please focus on inhibition zone diameters only
5-Please add the histogram in Fig.1.4
6-According to section 4.4, the authors prepared a series of Ag NPs inside the gum matrix but they characterized just one sample
7-The cytotoxicity test should be done please because the Ag NPs have known toxicity
8- Conclusion needs to be rewritten concisely and quantitively
9-The English of this manuscript should be revised again carefully
10-The references need to be updated
Comments on the Quality of English Language
The English of this manuscript should be revised again carefully
Author Response
Dear Dr. Ahmad,
Thank you again for your manuscript submission:
Manuscript ID: pharmaceuticals-3228584
Type of manuscript: Article
Title: Immobilization of silver nanoparticles with gum of Moringa oleifera
for fast and extremely effective antibacterial efficacy against highly
resistant bacterial species causing various infections in human
Please revise the manuscript found at the above link according to the
reviewers' comments and upload the revised file within 10 days.
Kind regards,
Ms. Celine Wan
E-Mail: celine.wan@mdpi.com
Respected Editor
Thank you very much for providing us with an opportunity to revise this manuscript. The editor and reviewers critically checked the whole manuscript and pointed out line-by-line mistakes. We have incorporated all the changes suggested by reviewers and tried our level best to enhance the quality of the manuscript. The Reviewers critically reviewed the manuscript and suggested very productive modifications, which will enhance the quality as well as data presentation.
We spent day and nights and revise this manuscript very carefully. The reviewers suggested a huge revision of grammatical and typographical mistakes. All the mistakes and errors have been removed carefully and revised the whole text according to the reviewer’s comments.
Hope that it will be considered for publication in your quality Journal.
Reviewer 1
The manuscript will be reviewed again after major revision.
Dear Professor,
Thank you very much for making effort to review our manuscript and for your nice suggestions. You have studied our manuscript very carefully and it is kind of you to spend your valuable time on reviewing our manuscript. We followed your nice comments and tried to improve our manuscript according to your suggestions. I am really happy to read your nice comments. It will really modify the current manuscript for the readers and for more clarity. Once again, I am really thankful for your critical comments. We have tried our level best to incorporate these comments/suggestions very carefully in the whole manuscript. Hope that its current form will be considered for publication.
My comments are:
Comment: 1-The idea is not completely novel. I see many Ag NPs immobilized on this gum, please define the novelty by the end of the introduction part and compare your results with other reported ones
Response: Good suggestions Sir. Most of the published articles from 2010 to 2024 used Gum arabica for encapsulation and as capping agent, however, the main objective of this study was to use a combo- efficiency of AgNPs with the defensive gum of Moringa plant. The moringa gum is naturally produce as a defensive substance to inhibit or stop the entrance of microorganisms to the plants. The Moringa gum possess antimicrobial potential but here we used both NPs and defensive gum in combination to enhance the activity against pathogenic microbes. Some authors used the moringa gum for the synthesis of NPs. If you suggest some relevant articles, we will must incorporate it in the MS.
Comment: 2- The title is too long, please concise it
Response: We have revised the title accordingly.
Comment: 3-Abstract should be revised again and rewritten concisely
Response: We have tried our level best to revise the abstract as “The worldwide misuse of antibiotics is one of the main factors in microbial resistance that is a serious threat to global health systems. Alternative strategies are needed to overcome this issue. In this study, a novel strategy was adopted to suppressed the growth of resistant pathogens. Here, an immobilization approach was applied using silver nanoparticles (AgNPs) and gum of medicinally important Moringa oleifera. The AgNPs were prepared from the leaves of Moringa oleifera and subsequently characterized through UV-spectrophotometry, FTIR, SEM and XRD. The differential ratios of characterized AgNPs were immobilized with Moringa gum. An immobilized AgNPs with gum of M. oleifera was investigated for antimicrobial potential against highly resistant pathogens. Immobilization is a powerful technique that is commonly used in pharmaceutics for controlled reactivity and emulsification. The immobilized AgNPs displayed promising activities against highly resistant B. subtilis (23.6 mm, 50 µL:200 µL), E. coli (19.3 mm; 75 µL:200 µL), K. pneumoniae (22 mm; 200 µL:200 µL), P. mirabilis (16.3 mm; 100 µL:200 µL), P. aeruginosa (22 mm; 175 µL:200 µL) and S. typhi (19.3; 25 µL: 200 µL) than either AgNPs alone or gum. The immobilized AgNPs released the positive sliver ions that is easily attached to the negatively charged bacterial cells. After attachment and permeation to bacterial cell, the immobilized NPs altering the cell membrane permeability, protein denaturation, causing oxidative stress, deactivating enzymes, generating ROS, DNA damage and changing gene expression level. It has been mechanistically considered that the immobilized AgNPs can kill bacteria by damaging their cell membranes, dephosphorylate tyrosine residues during their signal transduction pathways, cell apoptosis, rupture the organelles and inhibiting the cell division and finally lead to cell death. This study has a potential to be adopted by pharmaceutical industries as highly effective antibacterial alternatives to synthetic drugs.”
Comment: 4-E. coli (19.3; 75 µL:200) in the abstract should be changed to E. coli (19.3 mm; 75 µL:200), please focus on inhibition zone diameters only.
Response: Corrected as suggested.
Comment: 5-Please add the histogram in Fig.1.4
Response: Histogram has been added as suggested.
Comment: 6-According to section 4.4, the authors prepared a series of Ag NPs inside the gum matrix but they characterized just one sample
Response: Impressive comment. We only used those nanoparticles with best qualities including size and shape etc. for easy penetration to the bacterial cell.
Comment: 7-The cytotoxicity test should be done please because the Ag NPs have known toxicity
Response: Exactly respected Sir; this initial study and we are further working on it. But comparatively the Silver is traditionally used as antimicrobial agent from decades that is why we choose the greener synthesis of AgNPs as compared to other, while the gum minimizes the toxicity of the NPs. We also mentioned in the conclusion that further studies are needed to test its mechanism and its potential toxicity for environment and others.
Comment: 8- Conclusion needs to be rewritten concisely and quantitively
Response: We have revised the conclusion according to your nice suggestions as “The immobilized AgNPs displayed promising potential by inhibiting the growth of highly resistant B. subtilis, E. coli, K. pneumoniae, P. mirabilis, P. aeruginosa and S. typhi. These results concluded that the immobilized AgNPs have proven antimicrobial activities, however, the gum of M. oleifera may serve as a stabilizing or immobilizing agent for silver nanoparticles, enhancing their antibacterial effects. Silver nanoparticles are known to exhibit antimicrobial properties by interfering with bacterial cell membranes and other cellular structures, leading to the inhibition of bacterial growth. The immobilization strategy enhanced the antimicrobial efficacy that may be attributed to natural polymerization of gum and their stability and controlled release of silver ions. Further research and validation are needed to confirm the effectiveness, understand the underlying mechanisms, and assess the safety of such formulations for practical applications, especially in medical or environmental contexts.
”
Comment: 9-The English of this manuscript should be revised again carefully
Response: We have tried our level best to remove grammatical and language mistakes and errors from the revised MS accordingly.
Comment: 10-The references need to be updated
Response: We only mentioned the relevant and updated references. Kindly, if possible, suggest us updated relevant references other than, mentioned in the current MS for further incorporation.
Submitted for kind consideration.

Reviewer 3 Report
Comments and Suggestions for Authors
The manuscript entitled, " Immobilization of silver nanoparticles with gum of Moringa oleifera for fast and extremely effective antibacterial efficacy against highly resistant bacterial species causing various infections in human" by Ali et al. seems to overlap significantly with another article published earlier. My iThenticate determined 76% similarities.
However, the manuscript was not written as a scientific article rather it was written as a thesis report. In my opinion, the manuscript should be completely rewritten to be suitable for publication.
I have also an editorial comment on the number of coauthors of the manuscript. How can it be 18 for such a study?
The introduction, experimental section and results appear to be largely copied from other sources.
According to the study title, the total description is not focused and unnecessary literatures are included as references.
English grammar could be a little better. I have noticed throughout the manuscript several words and expressions that have meaningful translations but are not usual in scientific writing. Several sentences lack a pattern of punctuation (commas, periods, etc.). Thus, I recommend the article be processed by a native English speaker or any other scientific English expert.
The manuscript has been overloaded with many different data followed by a tiresome detailed description. The study seems to be a collection of separate experiments combined into a single report. It also seems to me a thesis report that needs to be written in the form of a manuscript.
The authors are advised to review and rethink the study problem and report only selected results corresponding to the major objectives. The reader cannot also find relevant conclusions that are related to the study hypotheses and the results verifying these assumptions.
Simply, the manuscript is too long, and it contains a collection of distantly related experimental works.
Bacterial names started with small letters such as: staphylococcus aureus, klebsiella pneumonia. The author used terms like "extremely effective", "miracle tree" sound subjective and should be avoided in scientific writing.
The statement "This study has a potential to be adopted by pharmaceutical industries" could be a better fit in the conclusion not in the abstract.
The introduction is too long and diverse. It should focus more precisely on the aim of the study rather than mixing other concepts like the anti-cancer, anti-diabetic, and anti-inflammatory properties of M. oleifera. While these points are interesting, they do not necessarily contribute to the central aim of the study.
There are some repetitions regarding the antimicrobial properties of M. oleifera and AgNPs, which could be efficient for better clarity.
The authors should focus more on Immobilization. More emphasis should be placed on explaining why immobilization is critical and how it enhances antibacterial efficacy. Also removing the redundant information and irrelevant properties of M. oleifera.
Table 1. 1: Nutritional composition of leaves, stem and roots of M. oleifera is not necessary for this study according to the title.
There are many figures which are not necessary for this study. Tab 3.1 and Fig 1.7. are same information.
How the use of UV-spectrophotometry, FTIR, XRD, and TEM analysis supporting the claims? Are these techniques required for this study?
The gum of M. oleifera were gathered from healthy and fresh plants but not mentioned how. The author should describe the sample collection in details for example, age of the plant, harvesting season, soil conditions, etc.
Several strains are selected for this study which is perfect but what are the rationales of selecting the strains?
The FTIR data is poorly organized and difficult to interpret also the XRD analysis seems incomplete.
The wide range of inhibition zones (for example, 11.6 mm to 23.6 mm for B. subtilis) indicates a lack of reproducibility.
Comments on the Quality of English LanguageEnglish language should be improved.
Author Response
Response letter to Editor and Reviewers comments
Dear Dr. Ahmad,
Thank you again for your manuscript submission:
Manuscript ID: pharmaceuticals-3228584
Type of manuscript: Article
Title: Immobilization of silver nanoparticles with gum of Moringa oleifera
for fast and extremely effective antibacterial efficacy against highly
resistant bacterial species causing various infections in human
Please revise the manuscript found at the above link according to the
reviewers' comments and upload the revised file within 10 days.
Kind regards,
Ms. Celine Wan
E-Mail: celine.wan@mdpi.com
Respected Editor
Thank you very much for providing us with an opportunity to revise this manuscript. The editor and reviewers critically checked the whole manuscript and pointed out line-by-line mistakes. We have incorporated all the changes suggested by reviewers and tried our level best to enhance the quality of the manuscript. The Reviewers critically reviewed the manuscript and suggested very productive modifications, which will enhance the quality as well as data presentation.
We spent day and nights and revise this manuscript very carefully. The reviewers suggested a huge revision of grammatical and typographical mistakes. All the mistakes and errors have been removed carefully and revised the whole text according to the reviewer’s comments.
Hope that it will be considered for publication in your quality Journal.
Reviewer 3
Comments and Suggestions for Authors
Dear Professor,
Thank you very much for making effort to review our manuscript and for your nice suggestions. You have studied our manuscript very carefully and it is kind of you to spend your valuable time on reviewing our manuscript. We followed your nice comments and tried to improve our manuscript according to your suggestions. Once again, I am really thankful for your critical comments. We have tried our level best to incorporate these comments/suggestions very carefully in the whole manuscript. Hope that its current form will be considered for publication.
Comment: The manuscript entitled, " Immobilization of silver nanoparticles with gum of Moringa oleifera for fast and extremely effective antibacterial efficacy against highly resistant bacterial species causing various infections in human" by Ali et al. seems to overlap significantly with another article published earlier. My iThenticate determined 76% similarities.
Response: Respected reviewer, every journal offers the option of uploading the MS as PRE-PRINT. The same option is also available in the Pharmaceuticals but I don’t click that option during the submission process. The earlier journal uploads the MS as pre-print. The Editor also ask the same question and I have sent the original reports (IA and Turnitin). The Editor then send the MS for review. It is our responsibility and we are well aware of the plagiarism, copyright and patents laws and regulations and its consequences.
Comment: However, the manuscript was not written as a scientific article rather it was written as a thesis report. In my opinion, the manuscript should be completely rewritten to be suitable for publication.
Response: Good suggestions but we have tried our level best to present our work with more clarity for broader readers as well as for scientific community and for reproducible experimentations. The table has been removed from the introduction.
Comment: I have also an editorial comment on the number of coauthors of the manuscript. How can it be 18 for such a study?
Response: Sure Sir. Please see the authors contribution. I do not see or read any restriction on the number of authors in the scope of the journal. If yes, kindly share with us for upcoming submissions to MDPI journals.
Comment: The introduction, experimental section and results appear to be largely copied from other sources.
Response: What is meant by “to be largely copied”. I think the reviewer is comparing the current MS with the same pre-print. The table has been removed from the introduction.
Comment: According to the study title, the total description is not focused and unnecessary literatures are included as references.
Response: We have tried to mentioned the relevant references from relevant articles. If you have objection; please highlight the irrelevant references for removal. We also removed the table from the introduction section.
Comment: English grammar could be a little better. I have noticed throughout the manuscript several words and expressions that have meaningful translations but are not usual in scientific writing. Several sentences lack a pattern of punctuation (commas, periods, etc.). Thus, I recommend the article be processed by a native English speaker or any other scientific English expert.
Response: The MS is already reviewed by native English expert before submission.
Comment: The manuscript has been overloaded with many different data followed by a tiresome detailed description. The study seems to be a collection of separate experiments combined into a single report. It also seems to me a thesis report that needs to be written in the form of a manuscript.
Response: The MS reported the synthesis of nanoparticles, its characterization, bacterial species isolation and identification and finally the antimicrobial activities. I think the MS having every detail needed for reproducible experiments.
Comment: The authors are advised to review and rethink the study problem and report only selected results corresponding to the major objectives. The reader cannot also find relevant conclusions that are related to the study hypotheses and the results verifying these assumptions.
Response: We have revised the abstract and the conclusion section accordingly.
Comment: Simply, the manuscript is too long, and it contains a collection of distantly related experimental works.
Response: We will upload some data/figures as supplementary materials with MS.
Comment: Bacterial names started with small letters such as: staphylococcus aureus, klebsiella pneumonia. The author used terms like "extremely effective", "miracle tree" sound subjective and should be avoided in scientific writing.
Response: The bacterial names have been corrected as suggested.
The term “extremely effective” has been removed according from the title as well as in the text as “Immobilization of silver nanoparticles with defensive gum of Moringa enhanced antibacterial efficacy against highly resistant bacterial species causing various infections in human”
Globally, the moringa plant is famous as miracle tree and published in peer-reviewed journals and books.
Comment: The statement "This study has a potential to be adopted by pharmaceutical industries" could be a better fit in the conclusion not in the abstract.
Response: We have removed it from the abstract and pasted it in the conclusion section accordingly.
Comment: The introduction is too long and diverse. It should focus more precisely on the aim of the study rather than mixing other concepts like the anti-cancer, anti-diabetic, and anti-inflammatory properties of M. oleifera. While these points are interesting, they do not necessarily contribute to the central aim of the study.
Response: Nice suggestion. Corrected as suggested. The table has been removed from the introduction and the introduction section has been reduced by removing unnecessary information’s.
Comment: There are some repetitions regarding the antimicrobial properties of M. oleifera and AgNPs, which could be efficient for better clarity.
Response: We have tried to remove these repetitions for more clarity from the whole text accordingly.
Comment: The authors should focus more on Immobilization. More emphasis should be placed on explaining why immobilization is critical and how it enhances antibacterial efficacy. Also removing the redundant information and irrelevant properties of M. oleifera.
Response: Strongly agree with the reviewer. We have removed the redundant information’s. Yes we explain it in introduction and discussion section.
Comment: Table 1. 1: Nutritional composition of leaves, stem and roots of M. oleifera is not necessary for this study according to the title.
Response: We have removed these information’s in the revised MS and only presents the relevant information’s.
Comment: There are many figures which are not necessary for this study. Tab 3.1 and Fig 1.7. are same information.
Response: Nice suggestions. The figure 1.7 will be transfer to the supplementary data.
Sensitivity pattern of B. subtilis, E. coli, K. pneumoniae, P. aeruginosa, P. mirabilis and S. typhi isolates were investigated using disk diffusion technique. Doxycycline (DO), Ceftriaxone (CRO), Cefotaxime (CTX), Sulbactam/cefoperazone (SCF), Erythromycin (E), Trimethoprim/sulfamethoxazole (SXT), Imipenem (IPM), and Azithromycin (AZM) have applied on each bacterial strain for susceptibility test as shown in Tab 3.1 and Fig 1.7 (supplementary data).
Comment: How the use of UV-spectrophotometry, FTIR, XRD, and TEM analysis supporting the claims? Are these techniques required for this study?
Response: We are strongly agreed with you. In the initial submission, we do not paste these information’s in the MS, BUT, the Editorial team send the MS for the incorporation of UV-Spectro, FTIR, XRD, and TEM. Let me discuss with the editor to move these information’s to the supplementary section, if necessary.
Comment: The gum of M. oleifera were gathered from healthy and fresh plants but not mentioned how. The author should describe the sample collection in details for example, age of the plant, harvesting season, soil conditions, etc.
Response: The details have been incorporated as short description “The samples were collected from three-year-old M. oleifera tree in March 2024 (Spring season), grown in clay soil with suitable soil pH (5.8) and temperature ranges from 28 to 32 ± 2 °C (Moring tree picture in supplementary file). These plants materials (gum and M. oleifera) were recognized and verified by Dr. Hina Fazal (a specialist in plant taxonomy) and the herbarium specimen with Voucher No. MBC-PES-10811 has been deposited in the herbarium of Medicinal Botanic Center (MBC), Pakistan Council of Scientific and Industrial Research (PCSIR), Laboratories Complex, Peshawar, Pakistan”
Comment: Several strains are selected for this study which is perfect but what are the rationales of selecting the strains?
Response: To test the efficacy of immobilized NPs against the most resistant strains suggested by the physicians in correlation with infections and antibiotics.
Comment: The FTIR data is poorly organized and difficult to interpret also the XRD analysis seems incomplete.
Response: We have made some addition and deletion for easy interpretation.
Comment: The wide range of inhibition zones (for example, 11.6 mm to 23.6 mm for B. subtilis) indicates a lack of reproducibility.
Response: We think and observed that the Bacillus species are more susceptible as compared to others and repeated the activity to confirm the results.
Submitted for kind consideration.

Round 2
Reviewer 1 Report
Comments and Suggestions for Authors
The “new” title “Immobilization of silver nanoparticles with defensive gum of Moringa oleifera for antibacterial efficacy against resistant bacterial species from human infections” is more adequate to the content of the manuscript.
The Abstract has clearly improved.
The keywords are now more adequate to the content of the study.
The introduction has been improved.
However, as previous advice, there is some information about Moringa oleifera that should be reduced to specific and related to antimicrobial properties.
The anti-inflammatory, anti-histaminic and anti-cholesterol properties of Moringa oleifera are not relevant for this study. The following sentences “M. oleifera bark extract has anti-inflammatory activity similar to diclofenac and thought to work by regulating the c-Jun N-terminal kinase and neutrophils pathway [24, 25], due to the presence of flavonoids, alkaloids, vanillin, moringin, phenols, tannins, carotenoids, hydroxymellein, Beta-sitosterol, and ß-sitostenol [26]. M. oleifera has been shown to have both fertility and anti-fertility properties. The aqueous extract of the plant has been found to more abortifacient and anti-fertility at doses of 200 and 400 mg/kg [15]. Recent study suggests that ingestion of M. oleifera before, after, and during pregnancy may lead to adverse fetal developmental outcomes [27]. The seed extract was able to reduce histamine release and suppress anaphylaxis induced by anti-immunoglobulin G. The mechanism underlying this effect is thought to be the membrane-stabilizing potential of the extract on mast cells [28]. The leaves extract can prevent gastric ulcers by reducing free radicals, neutralizing stomach acid, and increasing capillary resistance [29,30]. Some active constituents such as niazirmin A, niazirmin B, and niazimicin play a key role in lowering cholesterol levels and the avenasterol, stigmasterol, campesterol, and beta-sistosterol have diuretic activity [28,31]. The presence of quercetin, kaempferol, flavonoids, and benzylglucosinolate can protect the liver from damage [33,34]. The leaves, seeds, and dried pulp have been shown health inducing properties [28,35,36]. It is a nutrient-rich plant with a high protein content, as well as significant amounts of vitamins, minerals, fat carbohydrates, and dietary fibers [37-39].”should be removed from the manuscript
The results regarding Biogenic synthesis of nanoparticles and its characterization are correctly exposed.
The Discussion section is well structured and explains the results of the study.
The methodology has improved.
As previous said, the preparation of culture media (point 4.8) in not necessary. The authors just have to mention that Nutrient Agar preparation was performed following manufacturers indications.
The conclusions are sound and cautious considering the result of the study.
Author Response
Comment: The Abstract has clearly improved.
Response: Thank you Sir for your encouragement.
Comment: The keywords are now more adequate to the content of the study.
Response: Thank you.
Comment: The introduction has been improved.
Response: Thank you.
Comment: However, as previous advice, there is some information about Moringa oleifera that should be reduced to specific and related to antimicrobial properties.
The anti-inflammatory, anti-histaminic and anti-cholesterol properties of Moringa oleifera are not relevant for this study. The following sentences “M. oleifera bark extract has anti-inflammatory activity similar to diclofenac and thought to work by regulating the c-Jun N-terminal kinase and neutrophils pathway [24, 25], due to the presence of flavonoids, alkaloids, vanillin, moringin, phenols, tannins, carotenoids, hydroxymellein, Beta-sitosterol, and ß-sitostenol [26]. M. oleifera has been shown to have both fertility and anti-fertility properties. The aqueous extract of the plant has been found to more abortifacient and anti-fertility at doses of 200 and 400 mg/kg [15]. Recent study suggests that ingestion of M. oleifera before, after, and during pregnancy may lead to adverse fetal developmental outcomes [27]. The seed extract was able to reduce histamine release and suppress anaphylaxis induced by anti-immunoglobulin G. The mechanism underlying this effect is thought to be the membrane-stabilizing potential of the extract on mast cells [28]. The leaves extract can prevent gastric ulcers by reducing free radicals, neutralizing stomach acid, and increasing capillary resistance [29,30]. Some active constituents such as niazirmin A, niazirmin B, and niazimicin play a key role in lowering cholesterol levels and the avenasterol, stigmasterol, campesterol, and beta-sistosterol have diuretic activity [28,31]. The presence of quercetin, kaempferol, flavonoids, and benzylglucosinolate can protect the liver from damage [33,34]. The leaves, seeds, and dried pulp have been shown health inducing properties [28,35,36]. It is a nutrient-rich plant with a high protein content, as well as significant amounts of vitamins, minerals, fat carbohydrates, and dietary fibers [37-39].”should be removed from the manuscript
Response: We have removed the suggested information’s from the introduction.
Comment: The results regarding Biogenic synthesis of nanoparticles and its characterization are correctly exposed.
Response: Thank you.
Comment: The Discussion section is well structured and explains the results of the study.
Response: Thank you.
Comment: The methodology has improved.
Response: Thank you.
Comment: As previous said, the preparation of culture media (point 4.8) in not necessary. The authors just have to mention that Nutrient Agar preparation was performed following manufacturers indications.
Response: Thank you.
Comment: The conclusions are sound and cautious considering the result of the study.
Response: Thank you and already corrected.
Submitted for kind consideration
Reviewer 2 Report
Comments and Suggestions for Authors
No
Author Response
Thank you so much respected reviewer
Reviewer 3 Report
Comments and Suggestions for Authors
Thank you for the improvement of the manuscript and explanation point by point.
Comments on the Quality of English LanguageEnglish is fine to understand the study.
Author Response
Thank you for encouragement